# BTSP-CAM: A Brain-Inspired Geometric Memory for Class-Incremental Learning

Zheng Zhang [1]   Jiaye Yang [1]   Qingjie Guo [1]   Jiangrong Shen [2]   Long Chen [3]   Qi Xu [4]

## Abstract

Gradient-based optimization in class-incremental learning (CIL) often faces the plasticity–stability dilemma, as continuous weight updates can distort decision boundaries learned from earlier tasks. To address this issue, we revisit the problem from the perspective of stochastic geometric memory allocation and propose BTSP-CAM, a gradient-free memory algorithm grounded in theoretical insights from the hippocampal simpleBTSP model. Instead of fine-tuning a frozen encoder via backpropagation, BTSP-CAM externalizes plasticity into a binary synaptic matrix that evolves through local stochastic bit-flip updates. In particular, a trace-gated plateau process, driven by eligibility traces along with familiarity and collision signals, controls when and where synapses are rewritten, thereby suppressing cross-class interference in Hamming space. The resulting geometric memory states are mapped to semantic logits through a CA1-like competitive layer and a closed-form ridge readout, enabling fast consolidation after each task. Empirically, BTSP-CAM rivals gradient-based methods in a strictly exemplar-free setting and consistently boosts state-of-the-art baselines as a lightweight plugin. Finally, mechanistic analysis validates our geometric theory, confirming that stochastic repulsion actively bounds class overlap and stabilizes decision margins. Our code is available at https://github.com/ericzhengz/BTSP-CAM.

[1]Leicester International Institute, Dalian University of Technology [2]School of Computer Science and Technology, Xi'an Jiaotong University [3]Faculty of Medicine, Imperial College London [4]School of Computer Science and Technology, Dalian University of Technology. Correspondence to: Long Chen <lchen6@ic.ac.uk>, Qi Xu <xuqi@dlut.edu.cn>.

*Proceedings of the 43rd International Conference on Machine Learning*, Seoul, South Korea. PMLR 306, 2026. Copyright 2026 by the author(s).

## 1. Introduction

Deep neural networks achieve impressive performance when trained on fixed datasets, yet real-world applications rarely provide data in such a static form (Parisi et al., 2019; De Lange et al., 2019). Class-incremental learning (CIL) requires a model to learn new classes from a non-stationary data stream while retaining competence on previously learned ones (Rebuffi et al., 2017; Zhou et al., 2024c). The fundamental difficulty is the stability–plasticity dilemma: gradients updates that enable learning of new classes can simultaneously distort the decision geometry of previously learned ones, leading to catastrophic forgetting (French, 1999). Recent progress in pre-trained models (PTMs) appear to offer a way forward. Vision transformers (Dosovitskiy et al., 2021) and CLIP-style encoders (Radford et al., 2021) provide rich and transferable feature representations, allowing PTM-based CIL methods to freeze most backbone parameters and introduce lightweight adaptation modules (Wang et al., 2022a; Zhou et al., 2024b). Despite their success, these systems remain governed by gradient descent on continuous feature manifolds. As new tasks arrive, decision boundaries must be repeatedly adjusted within the same embedding space, and small updates can cumulatively erode the separation between old and new class manifolds.

Existing PTM-CIL approaches typically freeze the pretrained backbone and instantiate the plasticity-stability tension through various lightweight modules. Prompt-based methods, such as L2P and DualPrompt, employ a query-key matching mechanism to retrieve task-specific tokens from a shared pool to guide the frozen transformer (Wang et al., 2022b;a), while CODA-Prompt further enhances this via attention-based weighted summation of prompt components (Smith et al., 2023). Adapter-based strategies, including EASE, adopt a structural approach by expanding the network with lightweight task-specific branches and ensembling their outputs to mitigate interference (Zhou et al., 2024b; Sun et al., 2025b). Meanwhile, regularization- and distillation-based techniques constrain parameter updates to preserve old logits or features, often sacrificing plasticity to maintain stability (Li & Hoiem, 2016; Hou et al., 2019). Despite their architectural differences, these approaches share a common limitation: learning remains fundamentally driven

by gradient updates on continuous weights. As a result, forgetting is mitigated primarily through architectural bookkeeping mechanisms, such as buffers, distillation targets, or prompt pools, rather than through a fundamental change in how new information is written into memory.

In contrast, insights from the hippocampal literature offer a complementary perspective on continual learning (McClelland et al., 1995; O'Reilly et al., 2014). Behavioral Time-Scale Synaptic Plasticity (BTSP) was first identified in hippocampal CA1 as a plateau-potential gated plasticity mechanism that can induce place-field representations after single behavioral experience (Bittner et al., 2017). Subsequent studies demonstrated that BTSP can rapidly reshape existing representations through bidirectional synaptic changes, while mathematical analyses formalized this process in terms of seconds-long, eligibility-like traces coupled with sparse instructive events (Milstein et al., 2021; Cone & Shouval, 2021). Building on these biological rules, (Wu & Maass, 2025) proposed the simpleBTSP model, which instantiates BTSP-inspired, event-gated stochastic updates in a discrete setting, yielding a low-precision binary content-addressable memory. In this view, stochastic bit-flip dynamics in Hamming space give rise to both pattern completion and an LTD-like repulsion that limits cross-pattern overlap and stabilizes attractor structure (Wu & Maass, 2025).

Building on this perspective, we introduce BTSP-CAM, a zero-gradient associative memory framework that instantiates the theoretical principles of simpleBTSP within PTM-based CIL. First, at the algorithmic level, we externalize plasticity from the backbone into a binary synaptic matrix updated by trace-gated bit-flip rules, and couple it with a closed-form ridge readout so that class logits are consolidated from the evolving CA1 representation without backpropagation. Second, at the empirical level, we show that BTSP-CAM achieves competitive exemplar-free class-incremental performance on general benchmarks and consistently improves a range of PTM-based CIL baselines when used as a plug-in module. Third, at the mechanistic level, we analyze the learned memory geometry, covering CA1 drift, inter-class overlap, decision margins, and partial-cue robustness, and reveal how BTSP-style stochastic search realizes collision-aware repulsion that directly addresses the stability–plasticity dilemma in realistic continual learning.

## 2. Related Work

### 2.1. Class-Incremental Learning

A large body of CIL research mitigates forgetting by either replaying past information or constraining updates. Replay based methods store exemplars or generate pseudo-samples and often pair them with distillation to preserve previous outputs, as in Learning without Forgetting and derivative dis-

tillation pipelines(Li & Hoiem, 2016; Buzzega et al., 2020). Regularization and constraint based approaches instead limit harmful parameter changes, including importance-weighted penalties such as Elastic Weight Consolidation and gradient projection methods such as Gradient Episodic Memory(Kirkpatrick et al., 2017; Lopez-Paz & Ranzato, 2017). In addition, bias correction techniques address the recency induced imbalance that is typical in incremental training(Wu et al., 2019). These lines largely assume continual gradient optimization, which makes stability sensitive to optimization dynamics and often requires explicit access to past information or surrogate constraints.

Recent studies indicate that, with sufficiently strong representations, performance is increasingly determined by how the incremental decision boundary is updated rather than by re-learning features(Zhou et al., 2024a). This has renewed interest in replay-free heads that can be updated with sufficient statistics or closed-form solvers, including streaming LDA style classifiers and related analytic updates(Hayes & Kanan, 2020; Panos et al., 2023; Xue et al., 2026). Our work follows this replay-free and optimization-light direction, while focusing on a different stability mechanism by externalizing plasticity into a discrete memory substrate that resolves interference through local stochastic allocation(Shen et al., 2024; 2025).

### 2.2. Continual Learning with Pre-trained Models

In the pre-trained model based setting, many CIL methods aim to adapt with a small number of trainable components. Prompt based approaches learn task conditioned tokens and retrieval rules on top of a fixed Transformer, including L2P, DualPrompt, and CODA-Prompt(Wang et al., 2022b;a; Smith et al., 2023), with variants such as Domain-Adaptive Prompting improving robustness under shifts(Hong et al., 2024). Adapter based methods update lightweight modules or calibrated classifiers while keeping most parameters intact, as in SLCA and follow-up refinements(Zhang et al., 2023). Complementary work revisits the classifier layer itself and shows that random feature expansions or analytic heads can be competitive on strong embeddings, as in RanPAC and SimpleCIL(McDonnell et al., 2023; Zhou et al., 2024a). A recent survey provides a comprehensive taxonomy of this PTM-CIL landscape(Zhou et al., 2024d).

### 2.3. Associative Memory and BTSP-inspired Plasticity

External memory and content addressable retrieval have a long history in machine learning, spanning energy based associative memories such as dense and modern Hopfield networks(Ni et al., 2025), as well as differentiable memory augmented architectures such as Neural Turing Machines and Differentiable Neural Computers(Krotov & Hopfield, 2016; Ramsauer et al., 2020; Graves et al., 2014; Xu et al.,

2023). In neuroscience, hippocampal theories emphasize fast episodic storage and subsequent consolidation, and recent experiments identify Behavioral Time-scale Synaptic Plasticity as a mechanism for rapid association gated by dendritic plateau potentials(McClelland et al., 1995; Bittner et al., 2017; Milstein et al., 2021). The computational simpleBTSP model translates these principles into a discrete, low-precision CAM with stochastic potentiation and depression that can induce a repulsion effect for correlated patterns(Wu & Maass, 2025). BTSP-CAM builds on this line by instantiating simpleBTSP style stochastic plasticity as an explicit geometric memory core for exemplar-free CIL, connecting binary collision resolution in Hamming space to stable incremental decision making in the frozen PTM regime.

## 3. Preliminaries

### 3.1. PTM-based Class Incremental Learning

Class-incremental learning addresses the problem of learning knowledge from a sequential stream of disjoint tasks while retaining performance on previously learned classes (Rebuffi et al., 2017). Formally, let a sequence of $T$ tasks be denoted as $\{\mathcal{D}^1, \mathcal{D}^2, \ldots, \mathcal{D}^T\}$. Each task $\mathcal{D}^t = \{(\mathbf{x}_i, y_i)\}_{i=1}^{N_t}$ consists of $N_t$ training instances, where $\mathbf{x}_i \in \mathcal{X}$ represents the input data and $y_i \in \mathcal{Y}^t$ denotes the corresponding label. A fundamental constraint in this setting is that the label sets between different tasks are mutually disjoint, satisfying $\mathcal{Y}^i \cap \mathcal{Y}^j = \emptyset$ for all $i \neq j$. During the training phase of the $t$-th task, the learning algorithm is restricted to access data exclusively from the current dataset $\mathcal{D}^t$. This work adheres to the strict exemplar-free setting (Wang et al., 2022a; Zhou et al., 2024a), wherein the storage of historical samples from previous tasks $\mathcal{D}^{1:t-1}$ is prohibited. The primary objective is to construct a unified function $f : \mathcal{X} \to \mathcal{Y}_{seen}$ capable of minimizing the empirical risk over the union of all encountered classes $\mathcal{Y}_{seen} = \bigcup_{i=1}^{t} \mathcal{Y}^i$. Mathematically, this objective is formalized as identifying an optimal hypothesis $f^*$ within the hypothesis space $\mathcal{H}$ that minimizes the expected classification error across the cumulative test distribution. Denoting the testing set for the $i$-th task as $\mathcal{D}_{test}^i$, the optimization goal at stage $t$ is expressed as

$$f^* = \arg\min_{f \in \mathcal{H}} \mathbb{E}_{(\mathbf{x}, y) \sim \bigcup_{i=1}^{t} \mathcal{D}_{test}^i} \left[ \mathbb{I}(y \neq f(\mathbf{x})) \right] \quad (1)$$

where $\mathbb{I}(\cdot)$ denotes the indicator function, evaluating to 1 if the prediction is incorrect and 0 otherwise. An effective CIL model satisfying this criterion demonstrates discriminative abilities across all classes while balancing the acquisition of new concepts and the retention of previously acquired knowledge without catastrophic forgetting.

In the context of PTM-based continual learning, the predic-

tive model is structurally decoupled into two distinct components consisting of a pre-trained feature extractor $\phi(\cdot)$ and a downstream decision module $\mathcal{M}(\cdot)$. The formulation is expressed as $f(\mathbf{x}) = \mathcal{M}(\phi(\mathbf{x}))$. Here, $\phi : \mathcal{X} \to \mathbb{R}^d$ represents a deep neural network backbone pre-trained on a large-scale generic dataset. For a standard Vision Transformer (ViT) backbone(Dosovitskiy et al., 2021), the input image $\mathbf{x}$ is first tessellated into a sequence of patches and projected into linear embeddings. These embeddings are prepended with a learnable class token and processed through multiple layers of multi-head self-attention and multi-layer perceptrons. The final representation $\mathbf{h} \in \mathbb{R}^d$ is derived from the output state of the class token at the last transformer layer. A defining characteristic of the frozen PTM setting is that the parameters of $\phi$ remain static throughout the entire incremental learning process. Consequently, the continuous feature embedding $\mathbf{h} = \phi(\mathbf{x})$ is time-invariant with respect to the tasks. The adaptation to the evolving label space $\mathcal{Y}_{seen}$ relies entirely on the plasticity of the downstream module $\mathcal{M}$, which must map the fixed high-dimensional embedding $\mathbf{h}$ to task-specific decision boundaries. This constraint necessitates a downstream system capable of rapid pattern separation and association in the fixed latent space provided by the backbone.

### 3.2. The simpleBTSP Model

To address the challenge of rapid adaptation in fixed latent spaces without gradient updates, this work builds upon the theoretical framework of simpleBTSP(Wu & Maass, 2025). This model abstracts biological Behavioral Time-scale Synaptic Plasticity into a computational paradigm for maintaining high-capacity Content Addressable Memory (CAM) using low-precision components. Unlike traditional Hebbian learning rules that rely on millisecond-scale pre-post spike coincidence, simpleBTSP operates on behavioral time scales and constructs memory through binary interactions. The framework posits a synaptic weight matrix $W_B \in \{0, 1\}^{N \times M}$ connecting a high-dimensional input space to memory units, where both the input patterns $\mathbf{s} \in \{0, 1\}^M$ and the synaptic weights are constrained to be binary. This discretization significantly reduces the memory footprint and aligns with the hypothesis that hippocampal synapses operate as robust, low-precision switches rather than continuous variables.

A defining characteristic of this framework is that synaptic plasticity is not continuous but conditionally gated by a dendritic plateau potential $q_j \in \{0, 1\}$ in the post-synaptic neuron $j$. This potential serves as a binary instruction signal, indicating whether a specific memory slot should be updated, and its generation is stochastic, governed by a probability $f_q$ which typically depends on the novelty of the input pattern. This gating mechanism decouples the write command from the input activity. When a plateau potential is triggered

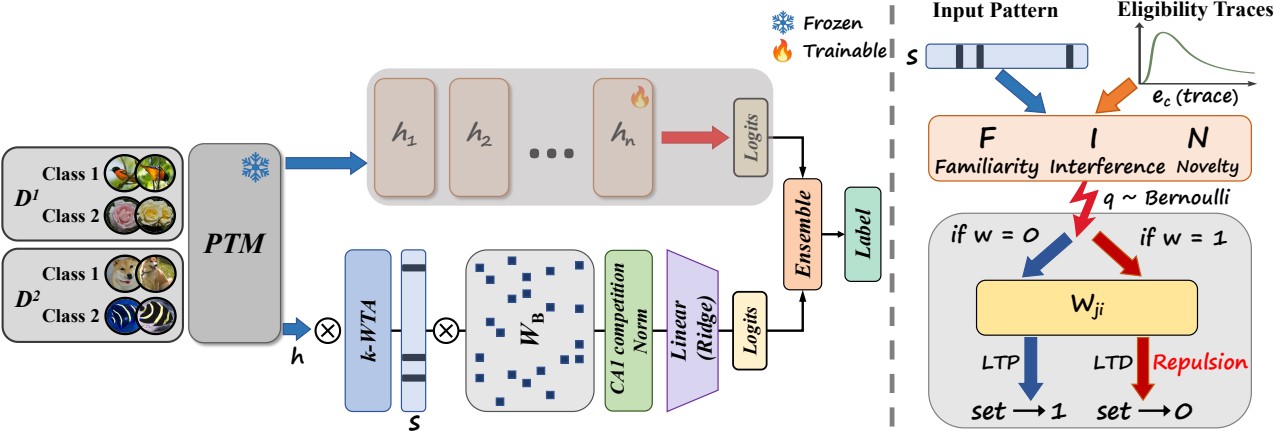

*Figure 1.* Illustration of BTSP-CAM. **Left**: Overall architecture, where a frozen PTM provides semantic features and a parallel BTSP-CAM branch converts them into sparse RF codes, stores them in a binary memory $W_B$, and produces CA1 logits via competition and ridge consolidation that can be ensembled with the PTM head. **Right**: Synaptic update mechanism, in which familiarity, interference, and novelty signals derived from the input code and eligibility traces gate BTSP-style bit flips, so that LTP ($0 \rightarrow 1$) allocates new patterns while collision-aware LTD ($1 \rightarrow 0$) repels conflicting ones in Hamming space.

($q_j = 1$), the synaptic update follows a stochastic bit-flip rule. Specifically, for any active presynaptic input ($s_i = 1$), the weight $w_{ji}$ is modified according to the following state-dependent transition:

$$w_{ji} \leftarrow \begin{cases} 1 & \text{if } w_{ji} = 0 \quad \text{(with probability } p) \\ 0 & \text{if } w_{ji} = 1 \quad \text{(with probability } p) \\ w_{ji} & \text{otherwise} \end{cases} \quad (2)$$

This concise formulation encapsulates two distinct plasticity mechanisms essential for continual learning: the transition from $0 \rightarrow 1$ (Long-Term Potentiation) facilitates the rapid, one-shot formation of memory attractors, while the transition from $1 \rightarrow 0$ (Long-Term Depression) stochastically eliminates weights active for multiple overlapping patterns. As demonstrated in (Wu & Maass, 2025), this LTD-driven mechanism induces a repulsion effect that actively orthogonalizes memory traces of correlated inputs, thereby enabling the system to separate highly similar patterns in high-dimensional spaces without requiring error-driven backpropagation.

## 4. Methodology

Facing the challenge of resisting catastrophic forgetting in PTM-CIL, we seek a mechanism that rectifies interference without continual backpropagation. The key idea of BTSP-CAM is to externalize plasticity into a discrete geometric memory core with two coupled steps: we map frozen backbone embeddings to sparse binary codes in Hamming space to make collisions explicit, and we use class-wise eligibility traces to gate plateau events so that stochastic bit-flip updates allocate new memories while repelling conflicts in $W_B$. The resulting CA1 representation is then consolidated

by a closed-form ridge readout to maintain stable margins with minimal optimization overhead.

### 4.1. Hamming Space Mapping via Sparse Coding

The theoretical foundation of simpleBTSP rests on the high capacity of binary synapses when inputs are sparse. To interface the continuous, dense manifold of the PTM with the discrete memory, we employ a fixed *Randomized Sparse Coding* layer. Given the backbone feature $\mathbf{h} = \phi(\mathbf{x}) \in \mathbb{R}^d$, we apply a high-dimensional random projection followed by a $k$-winner-take-all (k-WTA) nonlinearity(Johnson & Lindenstrauss, 1984):

$$\mathbf{u} = R\mathbf{h} + \mathbf{b}, \qquad \mathbf{s} = \mathbb{I}(\mathbf{u} \in \text{Top-}k(\mathbf{u})) \in \{0,1\}^m. \quad (3)$$

Here, $R \in \mathbb{R}^{m \times d}$ and $\mathbf{b} \in \mathbb{R}^m$ are drawn from standard normal distributions and frozen. By setting $m \gg d$ and maintaining a fixed sparsity ratio $k/m \ll 1$, this layer ensures that the probability of random collision between unrelated patterns is bounded. Since $\mathbf{s}$ is strictly binary, subsequent memory operations are reduced to efficient logical operations in Hamming space, providing a stable geometric canvas for stochastic allocation(Kanerva, 1988; Jiang et al., 2024; Li et al., 2024).

### 4.2. Adaptive Gating via Eligibility Traces

A defining feature of BTSP is that synaptic plasticity is not continuous but is gated by sparse, delayed instructive signals known as *Plateau Potentials*. In our CIL framework, we model this gating mechanism as an *adaptive importance sampling* process.

**Eligibility Traces as Temporal Context.** Synapses must bridge the gap between millisecond-scale input patterns and the seconds-scale plateau signals. We maintain class-specific eligibility traces $\mathbf{e}_c \in \mathbb{R}_{\geq 0}^m$ to track the geometric centroid of recent inputs. For an input $\mathbf{s}_t$ with label $y_t$, the traces evolve as:

$$\mathbf{e}_{c,t} = (1 - \lambda_{c,t})\mathbf{e}_{c,t-1} + \eta \cdot \mathbb{I}(c = y_t)\mathbf{s}_t, \qquad (4)$$

where $\lambda_{c,t}$ defines a state-dependent decay. This allows $\mathbf{e}_c$ to accumulate evidence for the current class while slowly forgetting inactive ones, providing a temporal context for the gating decision.

**Plateau Probability as Sampling Rate.** We define a scalar *plateau probability* $f_q^{\text{eff}}$ that determines the density of memory allocation. Unlike the fixed probability in biological models, we make $f_q^{\text{eff}}$ adaptive to the geometry of the current task. It is modulated by three signals:

- **Familiarity** $\mathcal{F} = \cos(\mathbf{s}, \mathbf{e}_y)$: Alignment with the current class trace.

- **Novelty** $\mathcal{N} = 1 - \mathcal{F}$: The geometric distance from the learned prototype.

- **Interference** $\mathcal{I} = \exp(-\beta \max_{k \neq y} \cos(\mathbf{s}, \mathbf{e}_k))$: A soft penalty for overlap with competing classes.

The effective gating probability is computed as:

$$f_q^{\text{eff}}(\mathbf{s}, y) = f_q^{\text{base}} \cdot \underbrace{4\mathcal{N}(1 - \mathcal{N})}_{\text{Geometric Prior}} \cdot \mathcal{F} \cdot \mathcal{I}. \qquad (5)$$

The term $4\mathcal{N}(1 - \mathcal{N})$ implements a geometric prior that favors the optimal learning zone, patterns that are novel enough to be informative but familiar enough to be consistent. The interference term $\mathcal{I}$ suppresses writing when the input is dangerously close to an existing rival class, preventing catastrophic collision. A binary plateau event $q \in \{0, 1\}$ is then sampled from Bernoulli($f_q^{\text{eff}}$).

### 4.3. Stochastic Search and Collision-Aware Repulsion

BTSP-CAM updates $W_B$ through a local stochastic search in discrete Hamming space. For each input $\mathbf{s}$, plateau events $q_j$ select a sparse subset of memory slots to update. For each input $\mathbf{s}$, we first sample slot-wise plateau indicators $q_j$ independently as $q_j \sim \text{Bernoulli}(f_q^{\text{eff}}(\mathbf{s}, y))$. Conditioned on $q_j = 1$, synaptic flips are sampled independently per synapse:

$$u_{ji} \sim \text{Bernoulli}(p), \qquad w_{ji} \leftarrow w_{ji} \oplus u_{ji},$$
$$\forall i \in \mathcal{A}(\mathbf{s}), \ \forall j \text{ with } q_j = 1, \qquad (6)$$

where $\mathcal{A}(\mathbf{s}) = \{i \mid s_i = 1\}$, and $w_{ji} \leftarrow w_{ji} \oplus u_{ji}$ is a compact notation for bidirectional BTSP with weight dependent direction: $u_{ji} = 1$ induces LTP ($0 \to 1$) if $w_{ji} = 0$,

---

**Algorithm 1** BTSP-CAM: Stochastic Geometric Search

1: **Input:** data stream $\mathcal{D}_t$, frozen backbone $\phi$, binary memory $W_B$, traces $\{\mathbf{e}_c\}$
2: **Hyperparameters:** $f_q^{\text{base}}, \beta, p, \eta, \{\lambda_{c,t}\}, \tau, \gamma$
3: **Phase 1: Stochastic Memory Allocation**
4: **for** each sample $(\mathbf{x}, y_t) \in \mathcal{D}_t$ **do**
5: $\quad \mathbf{s} \leftarrow \text{SparseCode}(\phi(\mathbf{x}))$ via Eq. (3)
6: $\quad$ Update traces $\{\mathbf{e}_c\}$ via Eq. (4)
7: $\quad f_q^{\text{eff}} \leftarrow f_q^{\text{eff}}(\mathbf{s}, y_t)$ via Eq. (5)
8: $\quad$ Sample $q_j \sim \text{Bernoulli}(f_q^{\text{eff}})$ for $j = 1, \ldots, S$ (independent over $j$)
9: $\quad \mathcal{A} \leftarrow \{i \mid s_i = 1\}$
10: $\quad$ Sample $u_{ji} \sim \text{Bernoulli}(p)$ for $i \in \mathcal{A}$ and $j$ with $q_j = 1$ (i.i.d. over $(j, i)$)
11: $\quad$ Update $w_{ji} \leftarrow w_{ji} \oplus u_{ji}$ for $i \in \mathcal{A}$ and $j$ with $q_j = 1$ via Eq. (6)
12: **end for**
13: **Phase 2: Systems Consolidation**
14: Construct CA1 features $Z_t$ via Eq. (7)
15: Update $(A_t, B_t)$ and compute $W_{\text{out}}^{(t)}$ via Eq. (8)

---

and LTD ($1 \to 0$) if $w_{ji} = 1$. The flip couples allocation and repulsion, since $0 \to 1$ supports attraction while $1 \to 0$ prunes shared bits under collisions.

### 4.4. Theoretical Properties

We explicitly connect our algorithm to the geometric properties derived in *simpleBTSP* theory. These properties explain why a binary stochastic system can outperform gradient-based fine-tuning in stability.

**Proposition 4.1** (Collision-Aware BTSP Bounds Class Overlap)**.** *Consider the overlap between two classes measured by the fraction of shared active bits in $W_B$. If plateau events are gated by collision signals and LTD flips occur with probability $p_{\text{LTD}} > 0$, then the expected cross-class overlap remains bounded by a constant $O(1)$ that does not grow with the number of tasks $T$.*

*Intuition.* Large overlap between classes increases the cosine similarity, which suppresses subsequent plateau probabilities via the $\mathcal{I}$ term. Simultaneously, the LTD mechanism stochastically flips shared bits to 0. This acts as a geometric regularizer, ensuring that new tasks carve out their own subspaces rather than overwriting existing ones.

**Proposition 4.2** (CA1 Readout Preserves Margins)**.** *Assume that the CA1 layer applies a competition operator and the ridge readout is re-estimated at the end of each task. Then, for classes that remain geometrically separated in the binary memory $W_B$, the decision margin under the linear readout shrinks at most linearly with the number of reused slots.*

*Intuition.* The normalization and thresholding in the CA1

*Table 1.* Comparison with state-of-the-art methods. The top section contains baselines, while the bottom section evaluates the effectiveness of our proposed BTSP-CAM plugin.

| Method | CIFAR B0 Inc5 | | CUB B0 Inc10 | | IN-R B0 Inc20 | | ObjNet B0 Inc10 | | OmniBench B0 Inc30 | | VTAB B0 Inc10 | |
| --- | --- | --- | --- | --- | --- | --- | --- | --- | --- | --- | --- | --- |
| | $\bar{\mathcal{A}}$ | $\mathcal{A}_B$ | $\bar{\mathcal{A}}$ | $\mathcal{A}_B$ | $\bar{\mathcal{A}}$ | $\mathcal{A}_B$ | $\bar{\mathcal{A}}$ | $\mathcal{A}_B$ | $\bar{\mathcal{A}}$ | $\mathcal{A}_B$ | $\bar{\mathcal{A}}$ | $\mathcal{A}_B$ |
| Finetune | 38.90 | 20.17 | 26.08 | 13.96 | 32.31 | 22.78 | 19.14 | 8.73 | 23.61 | 10.57 | 34.95 | 21.25 |
| Finetune Adapter | 60.51 | 49.32 | 66.84 | 52.99 | 58.17 | 52.39 | 50.22 | 35.95 | 62.32 | 50.53 | 48.91 | 45.12 |
| SimpleCIL | 87.57 | 81.26 | 92.20 | 86.73 | 61.26 | 54.55 | 65.45 | 53.59 | 79.34 | 73.15 | 85.99 | 84.38 |
| BTSP-only (ours) | 87.82 | 80.15 | 85.15 | 72.91 | 76.23 | 68.98 | 66.31 | 54.15 | 74.55 | 71.82 | 83.51 | 81.93 |
| L2P | 85.94 | 79.93 | 67.05 | 56.25 | 75.46 | 69.77 | 63.78 | 52.19 | 73.36 | 64.69 | 77.11 | 77.10 |
| + BTSP-CAM | 87.51 | 83.13 | 70.32 | 61.55 | 78.25 | 75.01 | 65.93 | 56.18 | 75.88 | 69.25 | 79.23 | 79.99 |
| DualPrompt | 87.87 | 81.15 | 77.47 | 66.54 | 73.10 | 67.18 | 59.27 | 49.33 | 73.92 | 65.52 | 83.36 | 81.23 |
| + BTSP-CAM | 88.93 | 84.08 | 79.81 | 70.85 | 76.72 | 73.45 | 61.45 | 53.81 | 76.09 | 69.91 | 85.11 | 84.55 |
| CODA-Prompt | 89.11 | 81.96 | 84.00 | 73.37 | 77.97 | 72.27 | 66.07 | 53.29 | 77.03 | 68.09 | 83.90 | 83.02 |
| + BTSP-CAM | 90.25 | 84.52 | 85.12 | 74.03 | 79.91 | 76.68 | 67.88 | 56.92 | 78.95 | 72.18 | 85.47 | 85.81 |
| APER+ Adapter | 90.65 | 85.15 | 92.21 | 86.73 | 75.82 | 67.95 | 67.18 | 55.24 | 80.75 | 74.37 | 85.95 | 84.35 |
| + BTSP-CAM | 91.82 | 87.91 | 92.83 | 87.05 | 77.35 | 70.62 | 68.51 | 58.15 | 81.98 | 77.83 | 87.02 | 86.99 |
| EASE | 91.51 | 85.80 | 92.23 | 86.81 | 81.74 | 76.17 | 70.84 | 57.86 | 81.11 | 74.85 | 93.61 | 93.55 |
| + BTSP-CAM | 92.33 | 86.95 | 92.71 | 87.13 | 82.88 | 78.15 | 71.95 | 60.59 | 82.03 | 77.91 | 94.15 | 94.52 |
| MOS | 93.30 | 89.25 | 93.49 | 90.12 | 82.96 | 77.93 | 74.69 | 63.62 | 85.91 | 80.05 | 92.62 | 92.79 |
| + BTSP-CAM | 93.85 | 90.12 | 93.82 | 90.47 | 83.79 | 79.18 | 75.38 | 65.81 | 86.45 | 82.55 | 93.08 | 93.81 |

layer (Eq. 7) ensure that only highly correlated memory slots contribute to the decision. Since LTD actively maintains separation in $W_B$, the CA1 layer provides a stable basis. The ridge regression then acts as a convex solver that optimally aligns this basis to the labels(Hoerl & Kennard, 1970), converting potential geometric drift into minor margin contraction rather than catastrophic boundary rotation.

### 4.5. Systems Consolidation via Ridge Readout

The final stage translates the evolved geometric state of the memory into semantic decision boundaries. First, we extract a geometry-regularized representation $\mathbf{z}(\mathbf{x})$ via a CA1-like competitive layer:

$$\mathbf{a}(\mathbf{x}) = \frac{W_B \mathbf{s}}{||\mathbf{s}||_1}, \qquad \mathbf{z}(\mathbf{x}) = \max(\mathbf{0}, \mathbf{a}(\mathbf{x}) - \tau). \quad (7)$$

This non-linearity suppresses noise and enforces sparsity in the retrieval. Then, we solve for the optimal readout weights $W_{\text{out}}$ using Ridge Regression:

$$A_t = A_{t-1} + Z_t^\top Z_t, \quad B_t = B_{t-1} + Z_t^\top Y_t,$$
$$W_{\text{out}}^{(t)} = (A_t + \gamma I)^{-1} B_t, \quad (8)$$

At the end of task $t$, we update sufficient statistics $(A_t, B_t)$ using only current-task data and compute $W_{\text{out}}^{(t)}$ in closed form, without storing any past exemplars.

$$\text{logits}_{\text{final}}(\mathbf{x}) = (1-\alpha) \cdot \text{logits}_{\text{PTM}}(\mathbf{x}) + \alpha \cdot (\mathbf{z}(\mathbf{x}) W_{\text{out}}^\top). \quad (9)$$

## 5. Experiment

This section presents a comprehensive empirical evaluation of our proposed **BTSP-CAM**. We first detail the experimental setup, including datasets, baselines, and evaluation protocols. We then present the main benchmark comparison, demonstrating the effectiveness of BTSP-CAM as both a standalone memory system and a versatile plugin for existing state-of-the-art methods. A detailed mechanistic analysis and ablation study will follow in Sec. 6.

### 5.1. Implementation Details

**Datasets.** We follow the exemplar-free PTM-CIL setting and evaluate on CIFAR-100 (Krizhevsky, 2009), CUB-200 (Wah et al., 2011), ImageNet-R (Hendrycks et al., 2021), ObjectNet (Barbu et al., 2019), OmniBenchmark (Zhang et al., 2022), and VTAB (Zhai et al., 2019). These datasets cover both standard CIL benchmarks and distribution-shifted evaluations.

**Dataset Split.** We adopt the standard 'B$m$-Inc$n$' notation for task splits(Rebuffi et al., 2017), where an initial task contains $m$ classes, followed by a sequence of tasks each introducing $n$ new classes. To ensure a fair and reproducible comparison, we use a fixed random seed of 1993 to shuffle the class order before partitioning the data. All methods are evaluated on the same sequence of training and testing sets(Zhou et al., 2024a).

**Training Details.** All experiments are implemented in PyTorch (Paszke et al., 2019) and PILOT (Sun et al., 2025a) on one NVIDIA RTX 4090 GPU. We employ a ViT-B/16, pre-trained on ImageNet-21K as the shared frozen backbone

for all methods. The BTSP-CAM module is configured with an RF dimension of 8192 and 4096 memory slots.

**Comparison Methods.** We benchmark against a suite of state-of-the-art PTM-based CIL methods, including prompt-based approaches (L2P(Wang et al., 2022b), DualPrompt(Wang et al., 2022a), CODA-Prompt(Smith et al., 2023)), adapter-based methods (APER(Zhou et al., 2024a), EASE(Zhou et al., 2024b), MOS(Sun et al., 2025b)). We also include SimpleCIL(Zhou et al., 2024a) as a strong non-parametric baseline. For completeness, a standard fine-tuning approach is reported as well.

**Evaluation Protocol.** Our evaluation follows the standard CIL protocol(Rebuffi et al., 2017). We report the Top-1 accuracy ($\mathcal{A}_B$) after the final task, and the average accuracy across all incremental stages, denoted as $\bar{A} = \frac{1}{B}\sum_{b=1}^{B}\mathcal{A}_b$.

## 5.2. Benchmark Comparison

Table 1 summarizes results on six benchmarks under two configurations: **BTSP-only** as a standalone gradient-free classifier, and **+BTSP-CAM** as a plugin augmenting existing methods. In **BTSP-only**, no external PTM-CIL host is used: the prediction is made solely by the BTSP-CAM branch on frozen features, thereby isolating the standalone memory predictor. In **+BTSP-CAM**, the host method is trained and evaluated with its original predictor and training procedure unchanged, while BTSP-CAM is attached as a parallel branch at the feature-to-decision interface and fused with the host logits following Eq. (9). BTSP-only is already competitive with several parameter-tuning baselines on multiple datasets, validating the core discrete memory architecture. When used as a plugin, BTSP-CAM yields consistent gains across diverse PTM-CIL methods, with particularly strong improvements in final accuracy $\mathcal{A}_B$, indicating reduced long-term forgetting. The gains are more pronounced on challenging or shifted benchmarks, highlighting the robustness of the proposed geometric allocation mechanism.

## 6. Mechanism and Analysis

In this section, we empirically validate the geometric mechanisms underlying **BTSP-CAM** derived from *simpleBTSP* (Wu & Maass, 2025). Using statistical logs and direct visualizations, we analyze four aspects: trace-gated alignment, LTD-driven repulsion, CA1-level structural stability, and associative robustness to partial cues.

### 6.1. Trace-Gated Plasticity as Geometric Alignment

We first examine how the eligibility traces guide the stochastic allocation process. In our framework, the trace $\mathbf{e}_c$ **acts not merely as a passive buffer, but as** a geometric refer-

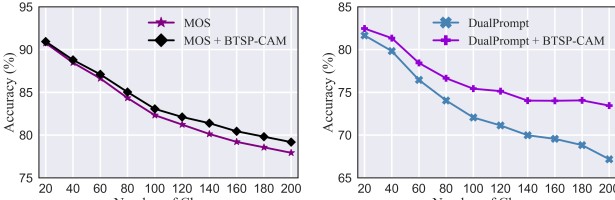

*Figure 2.* Performance comparison on ImageNet-R of BTSP-CAM as a plugin on MOS and DualPrompt.

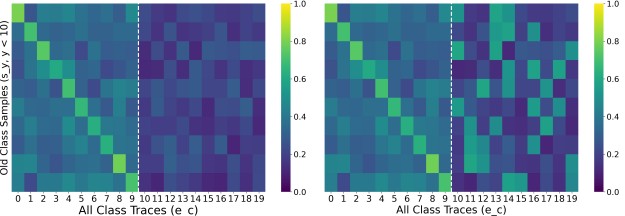

*Figure 3.* **Visualization of Geometric Repulsion.** Cosine similarity heatmaps between old-class samples (Task 1) and all memory traces. **(Left)** BTSP-CAM maintains an effective geometric exclusion zone, i.e., maintaining low similarity between old samples and new class traces. **(Right)** Without LTD, old samples erroneously activate new class traces (bright patches), showing **spurious activations** that cause interference.

ence vector that biases plasticity towards the existing manifold of a class.

Table 2 isolates the effect of trace-gated plateau probability $f_q^{\text{eff}}$. Removing gating increases forgetting (Fgt) and reduces geometric distinctiveness. More importantly, gating preserves a clear separation in familiarity between correctly classified samples and errors (higher $\mathbb{E}[F_c]$ than $\mathbb{E}[F_e]$), suggesting that the trace acts as a geometric filter that suppresses off-manifold writes and improves the signal-to-noise ratio of stored attractors.

*Table 2.* Effect of trace-gated plasticity. We report Average Accuracy ($\bar{\mathcal{A}}$), Forgetting (Fgt), mean familiarity for correct ($\mathbb{E}[F_c]$) and error ($\mathbb{E}[F_e]$) samples, and the number of plateau events per task (#Plat).

| Variant | $\bar{\mathcal{A}}$ (%) | Fgt (%) | $\mathbb{E}[F_c]$ | $\mathbb{E}[F_e]$ | #Plat. |
|---|---|---|---|---|---|
| BTSP-CAM | 86.82 | 14.33 | 0.45 | 0.32 | $\sim$550 |
| – trace gating | 85.07 | 17.05 | 0.38 | 0.36 | $\sim$820 |

### 6.2. Active Orthogonalization via Geometric Repulsion

A central tenet of the simpleBTSP theory is that LTD does not just forget old information, but actively repels confusing inputs to orthogonalize correlated memories(Yassa & Stark, 2011). We visualize this effect in **Figure 3** and quantify it in Table 3.

Figure 3 visualizes the cosine similarity between samples from old classes (Task 1) and memory traces of all tasks.

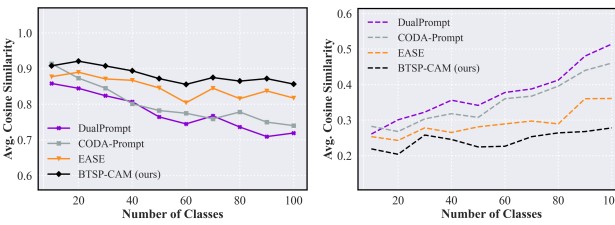

*(a) Within-Class Similarity*  *(b) Between-Class Similarity*

*Figure 4.* **Structural Stability Analysis. (a)** BTSP-CAM acts as a stable geometric anchor, maintaining high prototype consistency ($> 0.9$) compared to decaying baselines. **(b)** While baselines suffer from increasing overlap (drifting upwards), BTSP-CAM maintains low inter-class similarity, preserving distinct decision margins.

- **Without LTD (Right Panel):** We observe significant **spurious activations**—old samples strongly trigger the memory slots of newly learned classes (bright patches in the right block). This geometric collision implies that without repulsion, stochastic allocation blindly overwrites capacity, creating interference.

- **Full BTSP-CAM (Left Panel):** The interference block remains dark (deep purple). The system successfully creates a geometric exclusion zone, ensuring that new class representations remain orthogonal to old samples.

Table 3 quantifies this effect. Removing LTD **substantially increases** the Old-to-New error rate (from 8.5% to 20.3%) and raises inter-class overlap. This **supports the view** that LTD implements active orthogonalization: the stochastic search probabilistically prunes shared synapses, carving out separation in Hamming space without requiring negative data replay(Kanerva, 1988).

*Table 3.* Effect of collision-aware gating and LTD. We report Average Accuracy ($\bar{\mathcal{A}}$), Forgetting (Fgt), mean prototype overlap (Ov.), and Old-to-New error rate (O→N).

| Variant | $\bar{\mathcal{A}}$ (%) | Fgt (%) | Ov. | O→N (%) |
|---|---|---|---|---|
| BTSP-CAM | 86.82 | 14.33 | 0.18 | 8.5 |
| – collision term | 85.11 | 16.94 | 0.27 | 14.8 |
| – LTD (only LTP) | 83.67 | 18.81 | 0.32 | 20.3 |

### 6.3. Structural Stability and CA1 Geometry

The plasticity-stability dilemma in gradient-based CIL often manifests as feature drift. To contrast our stochastic allocation approach with gradient-based adaptation, we track the geometric evolution of memory prototypes in **Figure 4**.

As shown in Figure 4 (Left), gradient-based methods (DualPrompt, CODA-Prompt) exhibit a **characteristic decay** in prototype stability (dropping to $\sim 0.7$), as backpropagation continuously shears the feature space to accommodate

new tasks. **By contrast**, **BTSP-CAM acts as a stable geometric anchor**, with prototype similarity remaining above 0.9. Figure 4 (Right) further reveals that our method maintains a healthy margin between intra-class and inter-class similarities, whereas baselines show a collapsing margin.

Table 4 links this structural stability to the decision boundary. The CA1 competition normalizes and sparsifies the projected codes, while closed-form ridge consolidation refits the linear readout on top of the slowly drifting CA1 space. Removing these components exposes the raw memory to noise, degrading old-class accuracy and shrinking the margin ($1.25 \to 0.41$). This suggests that BTSP-CAM decouples plasticity from stability: plasticity is handled by local bit-flips, while stability is enforced by the frozen backbone and global consolidation.

*Table 4.* Effect of CA1 competition and ridge consolidation.

| Variant | $\bar{\mathcal{A}}$ (%) | Old (%) | Fgt (%) | Margin |
|---|---|---|---|---|
| BTSP-CAM | 86.82 | 83.15 | 14.33 | 1.25 |
| – CA1 competition | 82.34 | 74.92 | 20.17 | 0.68 |
| – consolidation | 75.81 | 63.18 | 28.59 | 0.41 |

### 6.4. Associative Robustness via Attractor Dynamics

Finally, we examine the associative memory behavior of the system, focusing on its ability to act as an attractor network that retrieves correct concepts from corrupted cues(Hopfield, 1982). To evaluate this, we randomly mask a fraction $p$ of the active bits in the sparse code **s**.

Table 5 shows that BTSP-CAM exhibits graceful degradation, retaining functional accuracy even when 60% of input bits are dropped ($p = 0.6$). This robustness **degrades much more rapidly** without CA1 competition or LTD. This experiment ties the PTM-CIL setting back to the neuroscience inspiration: the memory does not merely classify; it forms attractor basins. The stochastic geometry ensures that even partial patterns fall within the correct decision region, explaining the system's resilience to the distributional shifts inherent in continual learning.

*Table 5.* Partial-cue robustness. We report old-class accuracy (%) under different bit-drop ratios $p$.

| Variant | $p = 0.0$ | $p = 0.3$ | $p = 0.6$ | $p = 0.8$ |
|---|---|---|---|---|
| BTSP-CAM | 83.15 | 79.52 | 68.11 | 45.33 |
| – CA1 competition | 74.92 | 64.81 | 41.73 | 19.98 |
| – LTD | 78.33 | 69.57 | 50.91 | 26.14 |

## 7. Conclusion

This work establishes stochastic geometric allocation as a viable alternative to the prevailing gradient-based paradigm

for class-incremental learning. Our analysis demonstrates that externalizing plasticity into a binary framework governed by local rules effectively decouples memory retention from feature optimization. While the current formulation focuses only on image classification with frozen pre-trained backbones and predefined memory budgets, we find that biological mechanisms such as collision-aware depression act as essential inductive biases. They actively orthogonalize representations and enforce structural stability against the drift inherent in continuous streams. Ultimately, these results highlight that discrete stochastic search in Hamming space provides a mathematically grounded and computationally efficient path toward resilient continual learning. Looking forward, a promising direction is to develop continual learning systems that organize new knowledge through sparse, event-gated allocation and conflict-aware separation over robust representations.

## Impact Statement

This paper presents work whose goal is to advance the field of Machine Learning, specifically in the domain of bio-inspired Class-Incremental Learning. A primary advantage of our proposed BTSP-CAM framework is its strict exemplar-free setting, which eliminates the need to store historical data buffers, thereby reducing potential privacy risks associated with retaining user data. Additionally, by externalizing plasticity into a binary memory core and avoiding backbone backpropagation, our approach offers a pathway toward more energy-efficient edge learning systems.

However, we note that our method relies on frozen Pre-Trained Models (PTMs) as feature extractors. Consequently, any societal biases or fairness issues inherent in the pre-training data or the backbone architecture may be propagated to the downstream incremental tasks. While our geometric memory allocation does not explicitly amplify these biases, it does not actively correct them. Researchers and practitioners should remain aware of the underlying backbone's properties when deploying such systems in sensitive applications.

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

# A. Supplementary Material: Theoretical Derivations for BTSP-CAM

This supplementary material provides the rigorous mathematical foundations for the BTSP-CAM framework presented in the main paper. We formalize the geometric intuition of stochastic memory allocation and provide proofs for the key theoretical claims regarding stability and plasticity. The analysis is organized into four parts:

1. **Memory Allocation**: Analysis of collision probabilities in high-dimensional sparse Hamming space.

2. **Stochastic Plasticity**: Formal definition of the trace-gated bit-flip rule and derivation of one-shot learning probability.

3. **Pattern Separation**: Proof of Proposition 4.1, demonstrating that collision-aware LTD bounds inter-class overlap.

4. **Recall Stability**: Proof of Proposition 4.2, establishing the stability of decision margins and attractor dynamics under partial cues.

# B. Notation Glossary

*Table 6.* Summary of Mathematical Notation

| Symbol | Description |
|---|---|
| $\mathbf{x}$ | Input sample (e.g., image) |
| $\mathbf{h} = \phi(\mathbf{x})$ | Continuous feature embedding from a frozen PTM backbone ($\mathbf{h} \in \mathbb{R}^d$) |
| $\mathbf{s}$ | Sparse binary code in Hamming space ($\mathbf{s} \in \{0,1\}^m$, $\|\mathbf{s}\|_0 = k$) |
| $m, k$ | Dimension of sparse code and sparsity level (active bits) |
| $W_B$ | Binary synaptic memory matrix ($S \times m$) |
| $S$ | Number of memory slots |
| $q_j$ | Binary plateau potential indicator for slot $j$ |
| $f_q^{\text{eff}}$ | Effective gating probability for plateau potential (a function of novelty/familiarity) |
| $p$ | Probability of synaptic bit-flip (LTP/LTD) |
| $\mathbf{e}_c$ | Eligibility trace for class $c$ ($\mathbf{e}_c \in \mathbb{R}_{\geq 0}^m$) |
| $\mathcal{F}, \mathcal{N}, \mathcal{I}$ | Familiarity, Novelty, and Interference signals |
| $\tau$ | Retrieval threshold for CA1 competition layer |
| $\mathbf{z}(\mathbf{x})$ | Thresholded memory output (CA1 features) |
| $W_{\text{out}}$ | Linear readout weights learned via Ridge Regression |
| $\gamma$ | Regularization strength for Ridge Regression |
| $\alpha$ | Fusion weight for ensemble prediction |

## B.1. Core Assumptions

**Scope of Analysis.** The continual-learning stream is non-i.i.d. by construction. Our analysis does not model this macroscopic task process directly; instead, it characterizes the local collision geometry induced by the frozen backbone and the sparse coding map. Specifically, for unrelated concepts after coding, their active index sets are modeled as independent uniform $k$-subsets of $\{1, \ldots, m\}$, which makes the overlap geometry of the binary memory analytically tractable.

Our analysis relies on the following standard assumptions for modeling high-dimensional random data:

- **Localized Random-Code Model:** The independence assumption is used only at the coded-memory level. It does not assume an i.i.d. continual-learning stream, but models unrelated sparse codes as independent random active sets for the purpose of collision analysis.

- **Sparse Coding Regime**: We assume $k \ll m$, specifically $k = o(\sqrt{m})$, so that random collisions between unrelated sparse codes remain rare events.

# C. Theoretical Analysis of Memory Geometry

## C.1. Preliminaries and Notation

Let the input space be $\mathcal{X} \subseteq \mathbb{R}^d$ and the memory space be the Hamming space $\mathcal{H} = \{0, 1\}^m$, where $m \gg d$. We denote the mapping function as $\phi : \mathcal{X} \rightarrow \mathcal{H}$. The sparsity constraint is defined by $k \in \mathbb{N}$ such that for any $\mathbf{x} \in \mathcal{X}$, the binary code $\mathbf{s} = \phi(\mathbf{x})$ satisfies $\|\mathbf{s}\|_0 = k$. The sparsity ratio is given by $\xi = k/m$.

## C.2. Memory Allocation via Sparse Random Projection

The BTSP-CAM interface employs a randomized $k$-WTA coding scheme. Given an input embedding $\mathbf{h} \in \mathbb{R}^d$, the binary code $\mathbf{s}$ is generated via:

$$\mathbf{u} = \mathbf{W}_{in}\mathbf{h} + \mathbf{b}, \quad \mathbf{s} = \mathbb{I}(\mathbf{u} \in \text{Top-}k(\mathbf{u})) \tag{10}$$

where $\mathbf{W}_{in} \in \mathbb{R}^{m \times d}$ and $\mathbf{b} \in \mathbb{R}^m$ are initialized from $\mathcal{N}(0, 1)$. This process maps the continuous manifold to a discrete set of indices $\mathcal{I}(\mathbf{x}) = \{i \mid s_i = 1\} \subset \{1, \ldots, m\}$.

We analyze the collision probability between two distinct patterns to establish the geometric separation capability. Let $\mathbf{s}_a$ and $\mathbf{s}_b$ be binary codes derived from two uncorrelated inputs. We model their active indices $\mathcal{I}_a$ and $\mathcal{I}_b$ as independent uniform random subsets of size $k$ from the universe of $m$ indices. For the local collision analysis, we consider two unrelated coded patterns $\mathbf{s}_a$ and $\mathbf{s}_b$ after the frozen backbone and sparse coding map. Their active index sets $\mathcal{I}_a$ and $\mathcal{I}_b$ are modeled as independent unifor

**Lemma C.1** (Collision Probability in Sparse Hamming Space). *Let $\mathbf{s}_a, \mathbf{s}_b \in \{0, 1\}^m$ be independent random vectors with fixed sparsity $\|\mathbf{s}_a\|_0 = \|\mathbf{s}_b\|_0 = k$. The overlap magnitude $X = \mathbf{s}_a^\top \mathbf{s}_b$ follows a hypergeometric distribution. The probability that the overlap exceeds a threshold $t$ is bounded by:*

$$\Pr(X \geq t) \leq \exp\left(-m \cdot D_{KL}\left(\frac{t}{m} \,\|\, \frac{k^2}{m^2}\right)\right) \tag{11}$$

*where $D_{KL}(p \,\|\, q)$ is the Kullback-Leibler divergence between Bernoulli distributions with parameters $p$ and $q$.*

*Proof.* The overlap $X$ counts the number of shared active bits, which corresponds to the size of the intersection of two random sets $\mathcal{I}_a$ and $\mathcal{I}_b$. This is equivalent to drawing $k$ elements (indices of $\mathbf{s}_b$) from a population of $m$ elements, of which $k$ are successes (indices of $\mathbf{s}_a$). Thus $X \sim \text{Hypergeometric}(N = m, K = k, n = k)$.

The expected overlap is given by:

$$\mathbb{E}[X] = \frac{k \cdot k}{m} = m\xi^2 \tag{12}$$

For $m \gg k$, the hypergeometric distribution can be approximated by a binomial distribution $B(k, k/m)$. Applying the Chernoff bound for the hypergeometric tail probability, for any $t > \mathbb{E}[X]$:

$$\Pr(X \geq t) \leq \exp\left(-m\left(\frac{t}{m}\ln\frac{t/m}{\xi^2} + \left(1 - \frac{t}{m}\right)\ln\frac{1 - t/m}{1 - \xi^2}\right)\right) \tag{13}$$

The exponent term is exactly $-mD_{\text{KL}}(\frac{t}{m} \,\|\, \xi^2)$. $\qquad\square$

**Corollary C.2** (Capacity Bound). *Consider a sequence of $N$ stored patterns. To ensure that the maximum pairwise collision remains below a tolerance threshold $\tau$ with probability at least $1 - \delta$, the capacity $N$ must satisfy:*

$$N < \sqrt{\frac{2\delta}{\exp\left(-mD_{KL}\left(\frac{\tau}{m} \,\|\, \xi^2\right)\right)}} \tag{14}$$

*Proof.* Let $E_{ij}$ be the event that patterns $i$ and $j$ have overlap greater than $\tau$. By the union bound over all $\binom{N}{2}$ distinct pairs:

$$\Pr(\exists i < j : \mathbf{s}_i^\top \mathbf{s}_j \geq \tau) \leq \frac{N(N-1)}{2}\Pr(X \geq \tau) \tag{15}$$

We want this probability to be at most $\delta$. Substituting the bound from Lemma C.1:

$$\frac{N(N-1)}{2} \exp\left(-mD_{\mathrm{KL}}\left(\frac{\tau}{m} \parallel \xi^2\right)\right) \leq \delta \tag{16}$$

For $N \gg 1$, $N(N-1) \approx N^2$. Solving for $N$:

$$N^2 \leq 2\delta \cdot \exp\left(mD_{\mathrm{KL}}\left(\frac{\tau}{m} \parallel \xi^2\right)\right) \tag{17}$$

**Remark.** This collision bound depends on the spatial overlap of sparse active sets rather than the arrival order of tasks. It should therefore be understood as an order-agnostic local geometry estimate for the coded-memory layer, while the full non-i.i.d. continual-learning behavior is evaluated empirically in the main experiments.

Taking square roots yields the capacity bound. The exponential scaling with $m$ is due to the KL divergence term, which is positive when $\tau > m\alpha^2$ (i.e., when requiring overlap above chance level). $\qquad\square$

These derivations demonstrate that the high-dimensional sparse projection ensures rigorous pattern separation. Even with binary quantization, the probability of random collision decays exponentially with $m$. This geometric property provides the necessary condition for the subsequent stochastic synaptic updates to operate with minimal interference.

## D. Stochastic Bit-Flip Plasticity Rule

The BTSP-CAM model abstracts synaptic plasticity as a stochastic, event-driven process gated by a binary *plateau potential* signal $q_j \in \{0,1\}$. This signal controls the update of synapses onto memory slot $j$. The probability of triggering a plateau event, $f_q^{\mathrm{eff}}(\mathbf{s}, y)$, adapts dynamically based on the input pattern $\mathbf{s}$ and its label $y$, increasing for novel patterns and decreasing for familiar ones. Specifically, $q_j = 1$ indicates that memory slot $j$ is selected for storage. Given a plateau event ($q_j = 1$), the synapses $w_{ji}$ associated with active presynaptic inputs ($s_i = 1$) undergo a stochastic update defined by:

$$w_{ji}^{(t+1)} = \begin{cases} 1 & \text{with probability } p, \quad \text{if } w_{ji}^{(t)} = 0 \quad \text{(LTP)} \\ 0 & \text{with probability } p, \quad \text{if } w_{ji}^{(t)} = 1 \quad \text{(LTD)} \\ w_{ji}^{(t)} & \text{with probability } 1-p \quad \text{(No Change)} \end{cases} \tag{18}$$

where $p \in (0, 1]$ is the flip probability. This update only applies to indices $i$ where $s_i = 1$; otherwise, $w_{ji}^{(t+1)} = w_{ji}^{(t)}$.

This rule encapsulates two distinct plasticity mechanisms:

- **LTP (Allocation):** The transition $0 \to 1$ facilitates rapid memory formation. When a new pattern activates slot $j$, a subset of its active bits is stochastically potentiated, thereby encoding the pattern into the memory matrix $W_B$.

- **LTD (Repulsion):** The transition $1 \to 0$ provides a mechanism for erasing weights. If an input shares features with an existing memory trace in the same slot, the corresponding synapses may be depressed. This process effectively prunes connections that are active across multiple conflicting patterns.

### D.1. One-Shot Attractor Formation

The LTP component enables one-shot learning of new attractors. Consider a memory slot $j$ initially in the zero state, $\mathbf{w}_j = \mathbf{0}$. Upon presentation of a pattern $\mathbf{s}$ with $\|\mathbf{s}\|_0 = k$ and the triggering of a plateau potential $q_j = 1$, each of the $k$ active synapses has an independent probability $p$ of flipping to 1. The number of potentiated synapses, denoted by $K_{\mathrm{active}}$, follows a binomial distribution $B(k, p)$.

For reliable retrieval, the activation of the slot must exceed a threshold $\tau k$. We establish the probability of successful one-shot storage as follows:

**Lemma D.1** (One-Shot Storage Probability). *Let $\mathbf{w}_j^{(0)} = \mathbf{0}$. Given an input $\mathbf{s}$ with $k$ active bits and a plateau event $q_j = 1$, the probability that the resulting weight vector $\mathbf{w}_j^{(1)}$ activates above threshold $\tau k$ is:*

$$P(\mathbf{w}_j^{(1)} \cdot \mathbf{s} \geq \tau k) = 1 - F_{B(k,p)}(\lfloor \tau k \rfloor) \tag{19}$$

*where $F_{B(k,p)}$ is the cumulative distribution function of the binomial distribution. For $p > \tau$, this probability approaches 1 as $k \to \infty$. Specifically, applying the Chernoff bound yields:*

$$P(\mathbf{w}_j^{(1)} \cdot \mathbf{s} < \tau k) \leq \exp\left(-k D_{KL}(\tau \| p)\right) \tag{20}$$

*Proof.* The dot product $\mathbf{w}_j^{(1)} \cdot \mathbf{s}$ counts the number of successful $0 \to 1$ flips among the $k$ active bits, which is a random variable following a binomial distribution $B(k, p)$. The probability of this count being at least $\tau k$ is given by the survival function of the binomial distribution. For a non-integer threshold, the condition is equivalent to the count being greater than or equal to $\lceil \tau k \rceil$. The formula uses the floor function as it corresponds to the largest integer value strictly less than the threshold for the failure event. The result follows directly from the properties of the binomial distribution and standard concentration inequalities. □

This lemma confirms that with appropriate parameter choices ($p > \tau$ and sufficiently large $k$), a single update event creates a robust memory trace. Once stored, the high alignment between $\mathbf{s}$ and the updated $\mathbf{w}_j$ increases the familiarity signal, subsequently suppressing $f_q^{\text{eff}}$ and preventing redundant updates.

### D.2. Stochastic Search Dynamics

The update rule can be modeled as a local stochastic search in the weight space. For a given synapse $w_{ji}$ receiving constant active input ($s_i = 1$) across multiple plateau events, its state evolves as a two-state Markov chain. The transition matrix is given by:

$$M = \begin{pmatrix} 1-p & p \\ p & 1-p \end{pmatrix} \tag{21}$$

This symmetric process ensures that the stationary distribution is uniform over $\{0, 1\}$. However, in the context of BTSP-CAM, the adaptive gating mechanism breaks this symmetry temporally. A synapse is typically potentiated during the initial novelty phase (when $w_{ji} = 0$) and may be subsequently depressed only if it contributes to interference (collision) with a new, distinct pattern. This interplay between the Markovian update dynamics and the non-stationary gating signal allows the system to customize $W_B$ to the input distribution, balancing rapid acquisition with the active removal of interfering features.

## E. Pattern Separation via LTD: Bounded Overlap

A key property of the BTSP rule is its ability to induce a *repulsion effect* between memory traces of similar inputs. When two inputs share active features, the stochastic LTD ($1 \to 0$ transition) tends to eliminate these features from the shared memory representation, thereby orthogonalizing the traces in Hamming space. We formalize this mechanism by bounding the expected overlap between stored patterns.

Let $\mathbf{s}^{(a)}$ and $\mathbf{s}^{(b)}$ be two input patterns stored sequentially. Let $J$ be a memory slot involved in storing pattern $a$. Suppose pattern $b$ also triggers a plateau in slot $J$ ($q_J = 1$). We define the collision set $O_{ab}$ as the indices of synapses that are active after learning $a$ and receive active input from $b$:

$$O_{ab} = \{i \in \{1, \ldots, m\} \mid w_{Ji}^{(a)} = 1 \wedge s_i^{(b)} = 1\} \tag{22}$$

For each $i \in O_{ab}$, the update rule implies:

$$P(w_{Ji}^{(b)} = 0 \mid w_{Ji}^{(a)} = 1, s_i^{(b)} = 1) = p \tag{23}$$

Thus, each overlapping bit is eliminated with probability $p$ during the learning of $b$.

Consider a sequence of $T$ patterns $\mathbf{s}^{(1)}, \ldots, \mathbf{s}^{(T)}$. We analyze the survival of a specific synapse $w_{ji}$ across multiple storage events.

**Proposition E.1** (Bounded Expected Overlap per Synapse). *Fix a memory slot $j$ and a synapse $i$. Let $K$ be the number of distinct patterns in the sequence that activate synapse $i$ (i.e., $s_i^{(t)} = 1$) and trigger a plateau ($q_j^{(t)} = 1$). The expected number of patterns for which $w_{ji}$ remains continuously active is bounded by $1/p$. Specifically, the probability that $w_{ji}$ is shared by $k$ patterns decays geometrically:*

$$P(\textit{Shared by} \geq k \textit{ patterns}) = (1-p)^{k-1} \tag{24}$$

*Proof.* Assuming each plateau event is an independent trial, the event that synapse $i$ remains active after the storage of a new colliding pattern is a Bernoulli trial with success probability $1 - p$. The number of successful retentions, $X$, follows a geometric distribution with parameter $p$. The probability mass function is $P(X = k) = p(1-p)^{k-1}$ for $k \geq 1$. In practice, the gating mechanism introduces dependencies between trials (e.g., high familiarity suppresses future plateaus), which reduces the effective number of collision events. Therefore, this analysis provides a conservative upper bound on the expected overlap. The expected duration of activity (number of patterns) is given by the mean of the geometric distribution:

$$\mathbb{E}[X] = \sum_{k=1}^{\infty} k \cdot p(1-p)^{k-1} = \frac{1}{p} \tag{25}$$

Thus, a synapse active for pattern $a$ will, on average, survive only $1/p - 1$ subsequent collisions before being depressed. This value is independent of $T$. □

This proposition implies that no single synapse can serve a large number of patterns simultaneously. We now extend this to the overlap between entire memory traces. The overlap between two classes $a$ and $b$ is defined as the fraction of shared active synapses:

$$\mathcal{O}(a,b) = \frac{1}{S \cdot m} \sum_{j=1}^{S} \sum_{i=1}^{m} \mathbb{I}(w_{ji}^{(T)} = 1 \mid a, b \in \text{History}) \tag{26}$$

**Theorem E.2** (Bounded Memory Overlap). *Under the stochastic update rule with fixed $p \in (0, 1]$, the expected overlap between any two stored patterns $a$ and $b$ is bounded by a constant independent of $T$. Specifically:*

$$\mathbb{E}[\mathcal{O}(a,b)] \leq \beta(1-p) \tag{27}$$

*where $\beta = k/m$ is the sparsity ratio.*

pattern $a$ is stored first. Under the localized random-code model, the active support of a structurally unrelated coded pattern is treated as a uniform $k$-subset of $\{1, \ldots, m\}$. Thus, for a fixed presynaptic index $i$, the probability that $i$ belongs to the active support of $a$ is $\beta = k/m$. Conditioned on the selected memory slot and absorbing the constant LTP flip probability into the loose bound, the probability that $w_{ji}$ is set to 1 by $a$ is proportional to $\beta$

*Proof.* Consider a synapse $w_{ji}$. For it to contribute to the overlap, it must be active for both $a$ and $b$. Assume $a$ is stored first. The probability that $w_{ji}$ is set to 1 by $a$ is approximately $\beta$ (since each pattern activates a fraction $\beta = k/m$ of presynaptic indices).

For $w_{ji}$ to remain 1 after $b$ is stored, it must (i) be active in $b$ (a collision) and (ii) survive the LTD update during $b$'s storage. A conservative bound on the probability of joint activity is therefore:

$$P\big(w_{ji} \text{ active for both } a, b\big) \leq P(s_i^{(a)} = 1)\, P(s_i^{(b)} = 1)\, (1 - p) = \beta^2(1-p). \tag{28}$$

Equivalently, the expected number of shared bits after the update is upper bounded by the number of colliding active bits that survive LTD:

$$\mathbb{E}[\text{Shared Bits}] \leq k \cdot \Pr(s_i^{(b)} = 1) \cdot (1 - p) = k\xi(1-p). \tag{29}$$

Normalizing by $m$ yields a normalized overlap on the order of $\xi^2(1-p)$. Since $\xi \ll 1$, this overlap is small. For simplicity, we report the looser but always-valid upper bound

$$\mathbb{E}[\mathcal{O}(a,b)] \leq \xi(1-p), \tag{30}$$

which follows from $\mathcal{O}(a,b) \leq \xi$ and the survival factor $(1 - p)$ under collision. □

Theorem E.2 formalizes the intuition that BTSP's collision-aware updates prevent uncontrolled overlap. The repulsion mechanism ensures that whenever two patterns share features, those features are likely to be removed from the shared memory representation. Consequently, even highly correlated inputs result in nearly orthogonal traces in $W_B$. This repulsion effect, combined with the gating mechanism that suppresses plateau probabilities for high-interference inputs, provides a dual defense against catastrophic forgetting.

# F. Attractor Dynamics and Recall Stability

The previous section established that collision-aware LTD controls overlap in the binary memory matrix $W_B$. We now turn from memory-space separation to recall behavior. Specifically, we show that the CA1-style retrieval layer supports robust pattern completion from partial cues, and that the subsequent ridge readout preserves decision margins when the recalled representations remain geometrically separated.

We analyze the recall dynamics of the learned memory system, specifically its ability to function as an attractor network that retrieves correct patterns from partial or noisy cues. We also provide a formal justification for the stability of decision margins under the linear readout.

## F.1. Robust Pattern Completion

The recall process involves a feedforward pass $\mathbf{s} \to W_B\mathbf{s} \to \mathbf{z}(\mathbf{x})$ followed by a linear readout. The intermediate activation vector $\mathbf{a}(\mathbf{x}) \in [0,1]^S$ represents the normalized overlap:

$$a_j(\mathbf{x}) = \frac{\mathbf{w}_j^\top \mathbf{s}}{\|\mathbf{s}\|_1} \tag{31}$$

The CA1 layer applies a thresholding nonlinearity $\mathbf{z}(\mathbf{x}) = \max(0, \mathbf{a}(\mathbf{x}) - \tau)$. A non-zero $z_j(\mathbf{x})$ indicates that memory slot $j$ has successfully retrieved the pattern.

Consider a stored pattern $\mathbf{s}^*$ allocated to slot $j^*$. In the ideal case, $\mathbf{w}_{j^*} \approx \mathbf{s}^*$. When presented with a partial cue $\mathbf{s}'$ which retains $k_{\text{cue}}$ active bits from $\mathbf{s}^*$ (and no false bits), the activation is:

$$a_{j^*} = \frac{\mathbf{s}'^\top \mathbf{w}_{j^*}}{\|\mathbf{s}'\|_1} \approx \frac{k_{\text{cue}}}{k_{\text{cue}}} = 1 \tag{32}$$

Since $1 > \tau$, the slot $j^*$ fires. Conversely, for any unallocated slot $j \neq j^*$, the expected overlap is determined by the random collision rate $\alpha^2 m$. For sparse codes ($k \ll m$), this overlap is negligible:

$$\mathbb{E}[a_j] \approx \frac{k_{\text{cue}}\alpha}{k_{\text{cue}}} = \alpha \ll \tau \tag{33}$$

Thus, spurious slots remain silent. This demonstrates that stored patterns form **attractor basins**: input vectors within a certain Hamming radius are mapped to the correct memory state.

We quantify the robustness to noise. Let $\mathbf{s}'$ be a corrupted version of $\mathbf{s}^*$ with $e$ erasures (missing 1s) and $f$ false alarms (extra 1s). The activation condition for the correct slot is:

$$\frac{(k-e) - \epsilon_{\text{noise}}}{k - e + f} > \tau \tag{34}$$

For small perturbations $e, f \ll k$, the LHS approximates 1, ensuring robust retrieval. The threshold $\tau$ controls the trade-off between sensitivity and specificity.

## F.2. Proof of Proposition 4.2: Margin Preservation

We now formalize the claim that the linear readout preserves decision margins despite the geometric drift induced by continual learning.

**Proof of Proposition 4.2.** Let $\mathbf{z}_c$ be the ideal latent representation of class $c$. Due to LTD updates from subsequent tasks, the actual representation $\tilde{\mathbf{z}}_c$ degrades. We model this as $\tilde{\mathbf{z}}_c = \rho\mathbf{z}_c + \boldsymbol{\epsilon}$, where $\rho \in (0, 1]$ is the retention rate and $\boldsymbol{\epsilon}$ is sparse noise. The ridge regression solution for class $c$ is $\mathbf{w}_c = (\tilde{Z}^\top \tilde{Z} + \gamma I)^{-1}\tilde{\mathbf{z}}_c$. Assuming orthogonality between class manifolds (enforced by LTD), we approximate the Gram matrix $\tilde{Z}^\top \tilde{Z}$ as diagonal blocks. The projection of weights onto the signal direction is:

$$\mathbf{w}_c^\top \tilde{\mathbf{z}}_c \approx \frac{\|\tilde{\mathbf{z}}_c\|^2}{\|\tilde{\mathbf{z}}_c\|^2 + \gamma} = \frac{\rho^2\|\mathbf{z}_c\|^2}{\rho^2\|\mathbf{z}_c\|^2 + \gamma} \tag{35}$$

The decision margin $\Delta$ for a positive sample $\tilde{\mathbf{z}}_c$ against a negative class $k$ is:

$$\Delta = \mathbf{w}_c^\top \tilde{\mathbf{z}}_c - \mathbf{w}_k^\top \tilde{\mathbf{z}}_c \approx \frac{\rho^2\|\mathbf{z}_c\|^2}{\rho^2\|\mathbf{z}_c\|^2 + \gamma} - \delta \tag{36}$$

where $\delta$ is the bounded interference term (from Theorem E.2). Let $n_c(T)$ be the number of reused (colliding) slots for class $c$ after $T$ tasks. The retention rate $\rho$ decreases linearly with $n_c(T)$:

$$\rho(T) \approx 1 - \frac{n_c(T)}{k} \tag{37}$$

Substituting this into the margin equation, and performing a Taylor expansion for small degradation ($n_c \ll k$), we find that the margin $\Delta$ decreases linearly with $n_c(T)$.

$$\Delta(T) \approx \Delta(0) - \kappa \cdot n_c(T) \tag{38}$$

where $\kappa = \frac{\Delta(0)}{k} + \mathcal{O}(1/k^2)$ collects the first-order sensitivity of the margin to slot reuse under the approximation $\rho(T) \approx 1 - n_c(T)/k$. Since $n_c(T)$ grows at most linearly with $T$ under the bounded collision rate, the margin degradation is at most linear. Crucially, as long as the sparse regime ensures $n_c(T) \ll k$, the margin remains positive, preventing catastrophic forgetting. $\qquad\square$

