# OpenReview forum: "BTSP-CAM: A Brain-Inspired Geometric Memory for Class-Incremental Learning"
_ICML.cc/2026/Conference — ICML 2026 regular_

### Official Review · Reviewer_4QWM · 2026-03-07

**Soundness:** 3
**Presentation:** 4
**Significance:** 4
**Originality:** 2
**Overall Recommendation:** 4
**Confidence:** 4

**Summary:**

The paper proposed to stop using continual gradient updates to store new classes and instead puts plasticity into an external binary memory. It draws inspiration from the BTSP learning rule and the pattern separation and integration mechanisms in the CA1 region of the biological brain, and proposes a continual learning algorithm that prevents catastrophic forgetting caused by representation drift in pretrained models when learning new tasks. Essentially, it is a 'zero-gradient associative memory framework that instantiates the theoretical principles of simpleBTSP within PTbased CIL'. The claimed benefit is that this changes forgetting from “decision boundaries drifting in a continuous space” into “collision management in a discrete memory,”, or repulsion in the Hemming space, which the authors argue is easier to control.

**Compliance With Llm Reviewing Policy:**

Affirmed.

**Key Questions For Authors:**

1.	Many previous works have proposed algorithms based on brain-inspired sparse coding and model separation designs. However, the paper does not provide a detailed comparison with these related approaches. It would be helpful to expand the background section to discuss similarities and differences with existing methods and to include empirical comparisons of model performance.
2.	As the number of classification categories increases, how does the accuracy of the model change if the parameters of the BTSP_CAM component are kept fixed? Will performance degrade? If one wishes to maintain the same level of accuracy, how should the model size scale with the number of categories?
3.	Is the training of the BTSP_CAM component stable? Are there situations in which the model becomes difficult or impossible to train? It would be valuable if the authors could discuss the training stability of this module.
4.	Could the authors discuss the scalability of this algorithm to other types of tasks beyond classification, especially more diverse task settings?

**Limitations:**

No. The authors could discuss the limitations of the model in light of the questions raised above.

**Strengths And Weaknesses:**

The paper is clearly written, and the ablation studies are comprehensive. The model is thoroughly evaluated across multiple benchmarks. The authors also make a commendable effort to translate mechanisms from the biological brain into a concrete algorithm. One aspect not discussed in the paper, but worth considering, is the scalability of such an approach to settings with a much larger number of classes, to tasks beyond classification, and to models at larger scales.

Soundness: the results are clear and control experiments are performed accordingly.
Presentation: writing is clear and logical.
Significance: Continual learning is an important question. An efficient and discontinuous way to update memory is interesting.
Originality: Before this paper, PTM-based continual learning already had training-free / optimization-light heads, such as RanPAC’s frozen random projection plus class-prototype accumulation and gradient-based prompt/adapter methods. Also, the core biological inspiration—simpleBTSP as a binary one-shot content-addressable memory with BTSP-style updates—already existed as a published line of work by Wu and Maass. So this paper is fairly original as a synthesis, but not absolutely novel.

---

> ### Author Rebuttal · Authors · 2026-03-30
>
> Thank you for your valuable comments.
>
> **Q1 on the relation to previous works:**
> - Associative memory / sparse coding (e.g., Hopfield, SDM, simpleBTSP). We share the use of content-addressable or sparse memory. However, classical Hopfield-style memory is mainly a dense associative memory, SDM is a sparse distributed memory line, and simpleBTSP gives binary-synapse, one-shot associative memory[1]; BTSP-CAM differs in that it instantiates this memory logic in exemplar-free PTM-CIL, where frozen PTM features are first converted into binary codes and then updated by trace-gated writes with collision-aware LTD.
> - Model separation / capacity reservation (e.g., SpaceNet). We share the goal of reducing interference by separating plasticity from the full dense backbone. The difference is the mechanism: SpaceNet allocates sparse network capacity/free space[2], whereas BTSP-CAM places plasticity in an external binary memory branch with write/repulsion dynamics.
> - Frozen-backbone PTM-CIL (e.g., RanPAC, EASE). We share the frozen-pretrained setup, but differ in where adaptation lives: RanPAC relies on frozen random projections and class-prototype accumulation[3], and EASE uses task-specific lightweight adapters/subspaces; BTSP-CAM instead externalizes adaptation into a writable associative-memory system attached to frozen PTM features. We will revise the background to make these family-level similarities/differences explicit.
>
> **Q2+W1 regarding fixed-capacity scaling:**
> As the number of classes increases under fixed BTSP-CAM parameters, a graceful degradation in accuracy is theoretically expected due to increased shared-memory load and collision pressure within the bounded binary memory ($W_B$). Accordingly, Proposition 4.1 establishes a bounded-overlap and controlled-interference property for the sparse regime, rather than promising infinite capacity on a fixed budget. Importantly, the class-specific eligibility traces ($e_c$) act only as temporary routing signals, making the binary synaptic matrix the primary capacity constraint. To maintain high accuracy at scale, the system must dynamically preserve a comparable sparse operating regime. Our framework identifies two structural knobs to achieve this: expanding memory slots ($S$) and widening the code space ($m$) to keep collision rates controlled. Analogous to a biological synaptic memory sheet, storing a growing repertoire of memories inherently requires additional writable capacity. Thus, rather than claiming a precise asymptotic scaling law, we demonstrate a practical scaling principle: sustaining performance for larger class counts depends on explicitly scaling $S$ and $m$ to preserve sparse capacity and controlled interference.
>
> **Q3 On training stability:**  BTSP-CAM does not exhibit instability in the usual gradient-descent sense, since its plasticity is realized through local stochastic memory updates rather than deep end-to-end backpropagation. The practically relevant failure mode is therefore not failed optimization, but performance degradation under undersized memory budgets or excessive interference. Our ablations make this explicit: deliberately removing collision suppression or LTD increases mean inter-class overlap (0.18 -> 0.32) and old-to-new error bounds (8.5% -> 20.3%), while removing CA1 competition substantially lowers old-class accuracy (83.15% -> 63.18%). Under constrained budgets, the model therefore degrades through saturation/overlap, but remains trainable.
>
> **Q4 On scalability to diverse tasks:** The current empirical claim is deliberately scoped to exemplar-free class-incremental image classification with frozen pretrained encoders. Within that scope, BTSP-CAM is better viewed as a memory mechanism than as a classification-only trick: local BTSP-style writes and the synaptic matrix ($W_B$) operate on the geometry of frozen representations rather than on task-specific output heads. This suggests that the most transferable part is the external sparse-memory layer itself. What remains task-specific is the readout and supervision interface: extending to dense prediction, sequence generation, or other structured-output settings would require different decoders and supervision signals. The present paper therefore does not claim empirical validation on those tasks; the narrower point is that the proposed plasticity mechanism is compatible with broader frozen-representation pipelines, while broader task-level scaling remains open.
>
> **References:**\
> [1]. Trenton Bricken, et al. Sparse Distributed Memory is a Continual Learner. *arXiv preprint*, arXiv:2303.11934.\
> [2]. Ghada Sokar, et al. SpaceNet: Make Free Space for Continual Learning. *Neurocomputing*, 2021.\
> [3]. Mark D. McDonnell, et al. RanPAC: Random Projections and Pre-trained Models for Continual Learning. *NeurIPS*, 2023.

---

> > ### Author Rebuttal · Reviewer_4QWM · 2026-04-03
> >
> > The authors have fully addressed my concerns. I have no further question at this stage.

---

### Official Review · Reviewer_Nkm9 · 2026-03-12

**Soundness:** 3
**Presentation:** 3
**Significance:** 3
**Originality:** 3
**Overall Recommendation:** 5
**Confidence:** 4

**Summary:**

This paper proposes BTSP-CAM, a brain-inspired memory module for exemplar-free class-incremental learning (CIL). The approach shifts plasticity from continuously updated classifier weights to a binary associative memory that is updated through local stochastic bit flips, inspired by behavioral time-scale synaptic plasticity (BTSP).

The proposed framework operates as follows: A frozen pre-trained vision backbone first extracts feature representations from the input data. These features are then projected into sparse binary codes in Hamming space to enable efficient memory storage and retrieval. Next, a BTSP-inspired memory update rule uses stochastic bit flips corresponding to long-term potentiation (LTP) and long-term depression (LTD) to allocate memory for new classes while reducing interference with previously stored representations. During retrieval, a CA1-style competition mechanism selects the most relevant memory patterns. Finally, a closed-form ridge regression classifier maps the retrieved memory states to class logits for prediction.

Experiments on several benchmarks, including CIFAR-100, CUB-200, ImageNet-R, ObjectNet, OmniBenchmark, and VTAB, show that the BTSP-only approach achieves competitive performance in exemplar-free CIL. When used as a plugin, BTSP-CAM consistently improves strong PTM-based CIL methods such as L.

**Compliance With Llm Reviewing Policy:**

Affirmed.

**Final Justification:**

The authors delivered detailed responses that have completely resolved all the concerns raised in my initial review. On this basis, I have determined my final score.

**Key Questions For Authors:**

--How much improvement comes specifically from the BTSP update rule compared with a simpler binary memory head?
--Discuss whether the method can be extended to other backbones (e.g., CLIP) or to modalities beyond image classification.

**Limitations:**

Yes. The paper discusses limitations such as potential biases inherited from the pretrained backbone. However, it would also be helpful to explicitly mention that the experiments are currently limited to image classification with frozen pretrained models, and broader generalization remains an open question.

**Strengths And Weaknesses:**

Strengths:
--The paper addresses an important problem: exemplar-free continual learning with frozen pretrained models. The proposed module can act as a plugin that improves several strong baselines.
--The main novelty is the combination of BTSP-inspired memory updates with sparse binary representations and analytic classification. The work offers a different perspective on continual learning: stability via discrete memory allocation instead of gradient regularization.
--The method is technically reasonable and clearly defined. The algorithm pipeline is well described (feature extraction → sparse coding → BTSP updates → competition → ridge head). Experiments are conducted on multiple datasets and strong baselines.
--The paper is generally well structured and easy to follow. The motivation from neuroscience to machine learning is clearly explained. Figures help illustrate the memory architecture and update dynamics.

Weaknesses
--The theoretical analysis relies on simplified assumptions (e.g., independent inputs, sparse regimes) that may not fully match real pretrained feature distributions.
--Some claims could be more carefully phrased, especially regarding theoretical guarantees.
--Experiments focus mainly on image classification with a frozen ViT backbone, so broader applicability remains unclear.
--The distinction between BTSP-only and BTSP-CAM plugin results could be clearer.

---

> ### Author Rebuttal · Authors · 2026-03-30
>
> Thank you for your valuable comments.
>
> **W1 theoretical scope:** Our theoretical analysis (Props 4.1 & 4.2) is not intended as a macroscopic bound for arbitrary non-stationary CL streams. Rather, it provides a rigorous, mechanism-level guarantee exactly where our method operates: the coded-memory layer. It mathematically explains how sparse binary coding, combined with collision-aware BTSP updates, inherently bounds interference. Thus, our theory is a precise characterization of the discrete memory dynamics.
>
> **W2 settings:** Focusing on exemplar-free CIL is a widely recognized and pursued choice. It serves as the most rigorous regime to answer our central question: can plasticity be fully externalized from gradient-updated weights into a discrete memory? While extending this framework to dense prediction or multimodal generation would naturally require redesigning the task-level readout interfaces, the underlying BTSP associative memory rule remains entirely applicable. Our empirical scope precisely matches the core problem we set out to solve.
>
> **W3+Q2 dependence on ViT:** While our empirical scope focuses on exemplar-free image CIL, BTSP-CAM is fundamentally backbone-agnostic. It requires only a frozen feature stream, not a specific ViT architecture. We adopted the ViT-B/16-IN21K backbone in our main experiments strictly to ensure an apples-to-apples comparison with established PTM-CIL baselines. To test whether the memory mechanism transfers beyond frozen ViT features, we additionally ran BTSP-CAM on top of frozen **CLIP** image features on *CIFAR100 B0 Inc10*, using the recent **C3Box**[1], a CLIP-based CIL toolbox as the host implementation:
>
> | Method | Original Backbone | &nbsp;CLIP baseline $\bar{\mathcal A} / \mathcal A_B$ | &nbsp;&nbsp;+ BTSP-CAM $\bar{\mathcal A} / \mathcal A_B$ |
> | --- | --- | --- | --- |
> | DualPrompt | &nbsp;ViT-based | &nbsp;81.63 / 72.44 |&nbsp;&nbsp; 82.59 / 74.83 |
> | EASE | &nbsp;ViT-based | &nbsp;81.04 / 71.51 |&nbsp;&nbsp; 81.83 / 73.92 |
> | RAPF | &nbsp;CLIP-based | &nbsp;87.52 / 80.88 |&nbsp;&nbsp; 88.17 / 82.04 |
>
> DualPrompt and EASE are originally *ViT*-based methods that we transfer onto frozen CLIP image features, while RAPF[2] is already a *CLIP*-based host. For DualPrompt and EASE, BTSP-CAM is attached to the raw frozen CLIP image feature as a parallel image-memory branch, and the final fusion is performed at the class-logit level. For RAPF, BTSP-CAM is likewise driven by the raw frozen CLIP image feature, but fusion is applied after the image-text similarity head in class space.
>
> **W4 BTSP-only vs. Plugin:**
> - BTSP-only (Isolating the standalone memory branch): This setting evaluates whether the proposed discrete BTSP memory can act as a usable gradient-free predictor on frozen features by itself. It isolates the intrinsic discriminative utility of the memory branch, without relying on adaptation from an external PTM-CIL host.
> - BTSP-CAM plugin (Reducing residual continual-learning errors in a strong host): This setting evaluates whether the same memory mechanism can still improve a strong PTM-CIL host by correcting the part of continual-learning error that remains after host adaptation, especially cross-class interference, old-to-new confusions, and decision-margin erosion under a frozen representation stream. It should therefore be as a memory branch that mitigates residual forgetting-related errors at the final decision stage.
>
> **Q1 simpler binary memory head:**
> In the main paper, our ablations (-LTD, -CA1 competition, and -Consolidation) already isolate that removing these components effectively reduces BTSP-CAM to a passive binary scaffold. The resulting performance collapse proves that the gain is driven by BTSP’s active dynamics, not by binary storage alone.
> To provide microscopic evidence, we examined our early diagnostic logs of a simple-binary-memory head on *CIFAR 100 Inc-10, seed 1993*. We observed that certain classes rapidly became collision sinks that absorbed unrelated patterns. Specifically, after Task 3, class 35 emerged as a major sink: out of 500 diagnostic errors from older classes, class 3 was misclassified as class 35 twenty-two times, and the cluster of classes {0, 1, 3, 4} accounted for 45 total confusions into class 35. And this failure mode is precisely what the full BTSP-CAM prevents.
>
> **References:**\
> [1]. Hao Sun and Dawei Zhou. C3Box: A CLIP-based Class-Incremental Learning Toolbox. *arXiv preprint*, arXiv:2601.20852.\
> [2]. Linlan Huang, et al. Class-Incremental Learning with CLIP: Adaptive Representation Adjustment and Parameter Fusion. *ECCV*, 2024

---

> > ### Author Rebuttal · Reviewer_Nkm9 · 2026-04-03
> >
> > Thank you for the detailed and thoughtful rebuttal. I will maintain my original score 5.

---

### Official Review · Reviewer_LZw8 · 2026-03-12

**Soundness:** 2
**Presentation:** 2
**Significance:** 3
**Originality:** 3
**Overall Recommendation:** 4
**Confidence:** 3

**Summary:**

This paper presents BTSP-CAM. It operationalizes the biology inspired mechanism of Behavioral Time-scale Synaptic Plasticity for continual learning with a pretrained backbone. It uses randomized sparse coding on top of pretrained features, and uses that code to update a binary memory. This memory is updated so that classes that are different but have similar features have a larger hamming distance from each other, to minimize class confusion. The method can be used as a standalone gradient free classifier, or be added to existing methods. The paper shows that adding it onto existing methods results in consistent gains.

**Compliance With Llm Reviewing Policy:**

Affirmed.

**Final Justification:**

The rebuttal adequately answered my concerns.

**Key Questions For Authors:**

- Imagenet-A and Imagenet-R are listed as datasets used in the experiments, but seem to be missing from the main experiments table, why is that?
- Were the experiments run on multiple seeds? Do the results hold up if you do so?
- How were the hyperparameters tuned?
- How much extra memory does it take for this method? How does it scale with number of tasks? Prop 4.1 seems to imply that the method can scale infinitely with the number of tasks, but I think I might not be interpreting that correctly.
- One of the assumptions listed in the appendix for the theoretical arguments is that "Input Independence: Input samples $x_i$ are drawn independently and identically distributed (i.i.d.)." Isn't that violated by the setting of continual learning?

**Limitations:**

Yes

**Strengths And Weaknesses:**

Strengths:
- The paper’s core idea is interesting and novel. The use of sparse coding and BTSP to encode classes but repel the entries for similar classes to avoid confusion is intriguing.
- The empirical results are fairly strong. Bolting on BTSP onto other methods seems to universally lead to better performance.
- The ablations are interesting and help the understanding of why this method works.

Weaknesses:
- It seems as though the experiments only run one seed. For experiments of this scale, the experiments should run multiple seeds and the spread of the results should also be reported.
- There don't seem to be any details about hyperparameters and how they were tuned. What were the hyperparameters? Were they tuned on the same seed? Were other methods also tuned?
- Some of the theoretical arguments can be hard to follow.
- It is slightly unclear how BTSP-CAM integrates into other methods.

---

> ### Author Rebuttal · Authors · 2026-03-31
>
> Thank you for your valuable comments.
>
> **W1+Q2 multiple seeds:** We ran a 5-seed check (1993–1997) on *CIFAR100 B0 Inc10* with the ViT-B/16-IN21K backbone. The results are as follows:
>
> | Method | $\bar{\mathcal A}$ | $\mathcal A_B$ |
> |---|---:|---:|
> | DualPrompt | 86.42 ± 0.75 | 81.41 ± 1.00 |
> |&nbsp;&nbsp; + BTSP-CAM | 87.72 ± 0.68 | 84.39 ± 0.73 |
> | EASE | 91.76 ± 0.95 | 87.60 ± 0.28 |
> |&nbsp;&nbsp; + BTSP-CAM | 92.55 ± 0.84 | 88.59 ± 0.35 |
> | BTSP-only (ours) | 87.80 ± 0.66 | 81.09 ± 0.67 |
>
> **W2+Q3 hyperparameters:**
> BTSP-CAM introduces only a small, fixed memory-specific hyperparameter set: the RF dimension $m$, slot count $S$, sparsity level $k/m$, base write rate $f_q^{\mathrm{base}}$, and flip probability $p_{\mathrm{flip}}$, together with fixed trace-decay defaults. In the reported single-pass setting, we use $m=8192$, $S=4096$, $k\approx 164/8192$ ($2\\%$), $f_q^{\mathrm{base}}=0.15$, and $p_{\mathrm{flip}}=0.5$, with novelty and consistency modulation enabled. The host CIL baselines retain their *original* training settings, while the *same* BTSP-CAM configuration is used across baseline methods and datasets. In other words, we do not re-tune either the host baselines or the BTSP-CAM plugin on a per-seed, per-baseline, or per-dataset basis for the reported results.
>
> **W4 integration details:**
> BTSP-CAM is attached at the *feature-to-decision interface* of a host PTM-CIL method. The host method keeps its original predictor and training procedure unchanged. BTSP-CAM takes the host feature representation at this interface, maps it into sparse binary codes, performs BTSP-style memory write/retrieval, and converts the resulting memory state into auxiliary BTSP logits through its CA1/readout path. In plugin mode, the final prediction is obtained by late fusion between the host logits and the BTSP logits. In this sense, BTSP-CAM adds an external memory branch to the decision stage, rather than replacing the host method itself. We also report how to integrate BTSP-CAM into **CLIP-based** methods, you can see the rebuttal for **Reviewer Nkm9, W3+Q2 dependence on ViT.**
>
> **Q1 datasets using:** ImageNet-A was listed in error, we will delete it, while ImageNet-R is already included in Table 1 under the abbreviation “IN-R.”
>
> **Q4 extra memory and scale:**
> Proposition 4.1 is better interpreted as a *bounded-overlap / controlled-interference* statement in the sparse regime, rather than as a fixed-budget infinite-scaling claim. The extra memory introduced by BTSP-CAM is explicit and finite, dominated by the binary memory bank $W_B \in \\{0,1\\}^{S \times m}$ together with lightweight class-dependent statistics ($O(Cm)$) and readout weights ($O(CS)$). For the reported CIFAR100 setting ($S=4096$, $m=8192$, $C=100$), this is on the order of tens of MB, dominated by $W_B$ itself. With fixed $S$ and $m$, as the task stream or class set grows, the effective capacity per class decreases and collision pressure increases, so some performance degradation is expected unless the memory budget is increased. In this sense, Proposition 4.1 helps explain why interference remains controlled rather than unbounded, while the paper does not claim non-degrading performance under arbitrarily long task streams.
>
> **Q5 assumption of i.i.d:**
> The non-i.i.d. nature of continual learning is precisely why the supplementary material introduces the i.i.d. assumption only at a localized coded-memory level, after the frozen backbone and sparse coding map. Its role is not to model the full task stream, but to make the internal collision/overlap geometry induced by BTSP-CAM analytically tractable. In that local setting, the assumption supports the two proposition-level results: Proposition 4.1 on bounded expected cross-class overlap under collision-gated LTD, and Proposition 4.2 on controlled margin degradation under CA1 competition with ridge readout. Theoretical analysis and experiments therefore play different roles in the paper: the appendix analyzes the local memory mechanism that BTSP-CAM explicitly controls, while the empirical continual-learning results evaluate that mechanism on the actual non-i.i.d. task streams used in the paper.

---

> > ### Author Rebuttal · Reviewer_LZw8 · 2026-04-02
> >
> > Thank you for the rebuttal. I am raising my score to a 4, with the expectation that you will add multiple seeds for the rest of your experiments in the final version.

---

> > > ### Author Response · Authors · 2026-04-06
> > >
> > > Thank you again for your acknowledgement and for indicating that our rebuttal has adequately addressed your concerns.
> > >
> > > We just wanted to briefly follow up in case there are any remaining concerns that we may have missed. If not, we would be very grateful if your updated post-rebuttal assessment could also be reflected in the score/final justification when convenient. We confirm that multiple-seed results will be included in the final version appendix.
> > >
> > > Thank you again for your time and consideration.
> > >
> > > Best regards,\
> > > The authors

---

### Official Review · Reviewer_9b5s · 2026-03-13

**Soundness:** 2
**Presentation:** 3
**Significance:** 3
**Originality:** 3
**Overall Recommendation:** 4
**Confidence:** 5

**Summary:**

This manuscript proposes BTSP-CAM, a brain-inspired geometric memory mechanism for class-incremental learning, aiming to alleviate catastrophic forgetting by introducing a gradient-free binary synaptic memory evolving through stochastic bit-flip updates in Hamming space. The topic is interesting and relevant to continual learning.

**Compliance With Llm Reviewing Policy:**

Affirmed.

**Final Justification:**

Although the authors provided detailed responses, many of the concerns I raised in my initial review are difficult to fully resolve within the limited scope of the rebuttal. Based on this, I have made my final decision on the score.

**Key Questions For Authors:**

The geometric interpretation of forgetting and the move to discrete memory spaces is interesting and potentially promising. However, the current work remains largely empirical and heuristic, lacking sufficient theoretical grounding to establish a new learning paradigm. The questions are as follows:
1. The manuscript introduces an interesting perspective on continual learning by interpreting forgetting as decision geometry drift and proposing a binary geometric memory mechanism inspired by BTSP. This framing is conceptually appealing and provides a fresh angle compared to standard gradient-regularization or replay-based approaches. However, the methodological novelty could be more clearly articulated. While the proposed stochastic binary memory differs in implementation, it still follows the broader paradigm of freezing pretrained encoders and introducing lightweight adaptive decision modules, which have been explored in prior continual learning literature.
2. Biological inspiration serves mainly as conceptual motivation rather than a rigorously formalized model. Components such as eligibility traces, familiarity signals, and collision-aware updates are intuitively justified but lack deeper theoretical analysis regarding convergence behavior, stability conditions, or memory capacity. Strengthening the theoretical positioning would help clarify whether the approach represents a new learning principle or a well-designed heuristic mechanism.
3. Some aspects of the memory update dynamics could be described more systematically. Clearer exposition of how stochastic bit-flip probabilities are scheduled and how class allocation evolves over long task sequences would improve reproducibility and provide better insight into how decision boundaries are preserved.
4. The experimental results are generally encouraging and demonstrate consistent improvements across several class-incremental benchmarks, especially in plugin settings. Nevertheless, additional analysis on scalability to longer task streams, robustness under distribution shifts, and sensitivity to key hyperparameters would further strengthen the empirical case.
5. Another potential concern relates to the use of pretrained foundation encoders whose pretraining data may overlap semantically or categorically with the downstream evaluation benchmarks. In such cases, high classification accuracy may partially reflect inherited representational knowledge rather than the effectiveness of the proposed continual learning mechanism itself. This makes it difficult to isolate whether performance gains are due to improved resistance to forgetting or simply reduced task difficulty induced by strong prior features. Evaluating settings where pretrained models do not contain overlapping categories, or including stronger controls on representation transfer, would help clarify the contribution of the proposed memory dynamics.
6. Although a repository link is provided, it does not yet include runnable code. Providing full implementation would improve transparency and reproducibility, especially given the non-standard stochastic memory dynamics of the proposed approach.

**Limitations:**

A clearer discussion of limitations and potential failure cases is currently missing. Please provide it.

**Strengths And Weaknesses:**

Soundness:
The manuscript presents a reasonably well-specified methodological pipeline, including frozen pretrained feature extraction, sparse binary encoding into a Hamming space, stochastic Bernoulli bit-flip updates governed by eligibility traces and familiarity signals, and a competitive CA1-like readout combined with analytic ridge regression. The algorithmic flow is reproducible at a high level and supported by pseudocode and mechanism-oriented ablations. However, the overall technical rigor remains moderate rather than strong. Theoretical statements about collision control, margin stabilization, and robustness are provided only under restrictive assumptions and do not constitute full convergence, capacity, or generalization guarantees for the proposed stochastic discrete learning dynamics. In addition, biological inspiration is largely metaphorical and not formally grounded in computational neuroscience models, and important aspects such as sensitivity to memory size, sparsity level, or stochastic update rates are not thoroughly characterized.

Presentation:
The paper is clearly written and maintains a coherent narrative throughout. The central framing, interpreting continual forgetting as decision geometry drift and proposing discrete geometric memory allocation as a remedy, which is consistently articulated and supported by well-structured figures and empirical analyses. The progression from motivation to method, and from method to mechanism studies, is easy to follow, and the exposition avoids excessive formalism that might obscure the core idea. Some design choices, particularly the functional forms of plateau probabilities and competition rules, appear heuristic despite being presented in an intuitive and convincing storyline.

Significance:
The work targets exemplar-free class-incremental learning, a challenging and highly relevant setting, particularly in the era of large pretrained models where adaptation often shifts from representation learning to decision-level updates. Focusing on stabilizing classification geometry under frozen features is therefore a meaningful and timely research direction.

Originality:
The paper offers a relatively fresh geometric interpretation of continual forgetting and proposes a distinctive stochastic binary memory mechanism. Although many underlying components are known, their integration under a discrete geometric perspective gives the approach a recognizable identity. The contribution is best viewed as a novel system-level recombination rather than the introduction of a fundamentally new learning principle.

---

> ### Author Rebuttal · Authors · 2026-03-31
>
> Thank you for your valuable comments.
>
> **Q1 on novelty and methodological positioning:**
> The methodological novelty is the way continual plasticity is instantiated inside the frozen-PTM CIL regime: BTSP-CAM introduces a writable binary associative-memory layer with trace-gated updates, collision-aware LTD, CA1-style retrieval, and analytic consolidation to manage interference in decision geometry. This is the specific novelty we intend to articulate more clearly. At the same time, the method does follow the broader community paradigm of freezing pretrained encoders and adapting the decision stage. The intended point is therefore not to step outside PTM-based CIL, but to contribute a different plasticity mechanism within that now-standard framework.
>
> **Q2 On theoretical positioning:**
> The theory in this paper is centered on the sparse coded-memory geometry created on top of frozen representations. This is the object the method directly manipulates, so the analysis focuses on overlap accumulation, collision-aware LTD, and analytic consolidation, the quantities that determine the stability conditions and effective memory capacity of the proposed memory system; the corresponding empirical behavior is shown in **Q4**. Therefore, BTSP-CAM is best viewed as a *principled mechanism* with an explicit operating rule: BTSP/CA1 organize how memory is written, separated, and retrieved, and the paper instantiates that rule as a binary associative-memory system for exemplar-free frozen-PTM CIL.
>
> **Q3 on update dynamics and long-stream allocation:**
> The stochastic update has two levels. The write density is set by the adaptive plateau probability $f_q^{\mathrm{eff}}$, which starts from a base write rate and is modulated by familiarity/novelty, trace consistency, and cross-class collision pressure. Given an accepted plateau event, the synaptic update is a local bit-flip rule on the active dimensions of the sparse code: $p_{\mathrm{flip}}$ is fixed ($0.5$ in our reported setting), yielding BTSP-style LTP ($0 \to 1$) or LTD ($1 \to 0$).\
> Along long task streams, the memory budget stays bounded, so class allocation shifts from initial filling to reuse/pruning. Early writes occupy available support; later writes overwrite or prune conflicting support within the same budget. Decision-boundary preservation is then maintained by the same collision-aware mechanism: $f_q^{\mathrm{eff}}$ reduces unsafe writes, and LTD removes overlapping support across classes.
>
> **Q4 on empirical scope and sensitivity:**
> The current paper already provides extensive empirical validation under community-standard longer-stream CIL protocols (e.g., CIFAR100 B0 Inc5 and CUB B0 Inc10, featuring 20 incremental steps), as well as on shifted benchmarks (ImageNet-R, ObjectNet). These evaluations directly address the concerns regarding long-task scalability and robustness against distribution shifts. To directly characterize key hyperparameters, we add a compact local sensitivity check around *DualPrompt + BTSP-CAM* (CIFAR100 Inc10), with baseline $\bar{\mathcal A} / \mathcal A_B= 88.93 / 84.08$:
>
> | Factor | Setting | &nbsp;$\bar{\mathcal A} / \mathcal A_B$ |
> |---|---|---|
> | Memory slots \(S\) | 2048 | &nbsp;87.98 / 81.42 |
> |  | 4096 (default) | &nbsp;88.93 / 84.08 |
> |  | 8192 | &nbsp;**89.05 / 84.48** |
> | Active bits \(k\) | 64/8192 | &nbsp;88.08 / 81.74 |
> |  | 164/8192 (default) | &nbsp;**88.93 / 84.08** |
> |  | 328/8192 | &nbsp;87.58 / 81.36 |
> | Base write rate $f_q^{base}$ | 0.05 | &nbsp;88.00 / 81.86 |
> |  | 0.15 (default) | &nbsp;**88.93 / 84.08** |
> |  | 0.30 | &nbsp;87.64 / 81.72 |
>
> These results indicate a usable operating range around the reported default: enlarging memory yields diminishing returns, while overly sparse/dense codes or overly aggressive write rates mainly hurt final retention.
>
> **Q5 overlap concern:**
> The insight regarding potential semantic overlap touches upon a fundamental attribution challenge inherent to the broader frozen-PTM CIL paradigm. To rigorously navigate this within established community standards [1], the empirical evaluation employs a strictly controlled setup: all compared methods operate on the same backbone under identical incremental protocols. As shown in Table 1, *Finetune* uses the same backbone but no dedicated continual-learning mechanism, and it remains substantially weaker than BTSP-CAM and other CIL methods. Consequently, the empirical claims in this paper are strictly comparative: with the representation prior held constant, the observed improvements are demonstrably driven by the memory dynamics rather than reduced task difficulty. While introducing stricter low-overlap pretraining controls would provide valuable boundary insights for future research, the current controlled design securely validates the relative effectiveness of the continual adaptation mechanism reported here.
>
> **References:**\
> [1]. Dawei Zhou, et al. Continual Learning with Pre-Trained Models: A Survey. *IJCAI*, 2024.

---

> > ### Author Rebuttal · Reviewer_9b5s · 2026-04-06
> >
> > Thank you to the authors for the detailed response. However, I believe that the limitations of the paper were already clearly articulated in my initial review, and they are difficult to adequately address within a brief rebuttal. Therefore, I will maintain my original score.

---

### Decision · Program_Chairs · 2026-04-30

**Decision:**

Accept (regular)

**Comment:**

This paper studies exemplar-free class-incremental learning with frozen pretrained models and proposes BTSP-CAM, a brain-inspired binary associative memory mechanism for mitigating forgetting. Overall, the reviewers found the problem important and timely, the method technically interesting, and the empirical results strong, especially the consistent gains when BTSP-CAM is used as a plugin on top of competitive baselines. The paper was also generally considered clear and well presented.

The main concerns were about the limited scope of the theoretical analysis, the fact that the novelty is better characterized as a strong system-level synthesis rather than a fundamentally new learning principle, and some initial questions regarding experimental completeness, including seed robustness, hyperparameter details, and scalability.

In the rebuttal, the authors provided useful clarifications and additional evidence on these points, including multi-seed results, hyperparameter settings, integration details, and discussion of memory scaling and transfer beyond the main backbone setting. Most reviewers indicated that their concerns were adequately addressed after the rebuttal, while one reviewer remained somewhat cautious but still positive overall.

Taking the reviews and rebuttal together, my assessment is positive. The paper makes a solid contribution to continual learning with pretrained models and is suitable for inclusion in the ICML program. At the same time, the final version should better calibrate the theoretical claims and more clearly discuss the method’s current scope and limitations. My recommendation is Accept.